# The Neurodevelopmental and Molecular Landscape of Medulloblastoma Subgroups: Current Targets and the Potential for Combined Therapies

**DOI:** 10.3390/cancers15153889

**Published:** 2023-07-30

**Authors:** Hasan Slika, Paolo Alimonti, Divyaansh Raj, Chad Caraway, Safwan Alomari, Eric M. Jackson, Betty Tyler

**Affiliations:** 1Faculty of Medicine, American University of Beirut, Beirut P.O. Box 11-0236, Lebanon; hgs09@mail.aub.edu; 2Department of Neurosurgery, Johns Hopkins University School of Medicine, Baltimore, MD 21287, USA; draj3@jhmi.edu (D.R.); ccarawa2@jhu.edu (C.C.); salomar1@jhmi.edu (S.A.); ejackson@jhmi.edu (E.M.J.); 3School of Medicine, Vita-Salute San Raffaele University, 20132 Milan, Italy; p.alimonti@studenti.unisr.it

**Keywords:** medulloblastoma subgroups, molecular pathways, targeted therapy, combination therapy, neurodevelopmental origin, pediatric brain tumors

## Abstract

**Simple Summary:**

Medulloblastoma is the most common malignant brain tumor in the pediatric population. Despite the utilization of aggressive treatment modalities, including surgery, chemotherapy, and radiation therapy, patients with medulloblastoma still have a poor prognosis. Moreover, these modalities are associated with dramatic life-long complications. Hence, this calls for the development of novel therapeutic agents that can more effectively and safely target this tumor and improve the survival and quality of life for patients. The molecular-based classification of medulloblastoma into WNT activated, SHH activated, group 3, and group 4 opened the door for research endeavors that aim to study the specific cellular, molecular, and neurodevelopmental characteristics of each subtype. This review aims to summarize the literature on the different profiles of these subtypes, elaborate on the pharmacologic therapies that have been investigated to target each, and suggest potential combination therapies that can offer superior outcomes.

**Abstract:**

Medulloblastoma is the most common malignant pediatric brain tumor and is associated with significant morbidity and mortality in the pediatric population. Despite the use of multiple therapeutic approaches consisting of surgical resection, craniospinal irradiation, and multiagent chemotherapy, the prognosis of many patients with medulloblastoma remains dismal. Additionally, the high doses of radiation and the chemotherapeutic agents used are associated with significant short- and long-term complications and adverse effects, most notably neurocognitive delay. Hence, there is an urgent need for the development and clinical integration of targeted treatment regimens with greater efficacy and superior safety profiles. Since the adoption of the molecular-based classification of medulloblastoma into wingless (WNT) activated, sonic hedgehog (SHH) activated, group 3, and group 4, research efforts have been directed towards unraveling the genetic, epigenetic, transcriptomic, and proteomic profiles of each subtype. This review aims to delineate the progress that has been made in characterizing the neurodevelopmental and molecular features of each medulloblastoma subtype. It further delves into the implications that these characteristics have on the development of subgroup-specific targeted therapeutic agents. Furthermore, it highlights potential future avenues for combining multiple agents or strategies in order to obtain augmented effects and evade the development of treatment resistance in tumors.

## 1. Introduction

Medulloblastoma is defined as a WHO grade IV embryonal tumor that arises in the cerebellum or brain stem. It accounts for approximately 63.3% of intracranial embryonal tumors and approximately 20% of all pediatric brain tumors. It has a peak incidence between ages 0 and 9 years and exhibits a male predominance with a 1.7:1 male to female ratio [1]. The current mainstay of therapy for medulloblastoma is maximal safe surgical resection followed by risk-adapted craniospinal irradiation, a radiation boost to the primary tumor bed, and adjuvant multi-agent chemotherapy [2]. However, treatment with radiation therapy may be deferred in infants and toddlers less than 3 years of age due to the debilitating long-term neurocognitive effects of early exposure to radiation [3]. Unfortunately, despite this aggressive combination of treatment modalities, the 10-year survival rate of medulloblastoma remains less than 65% [1]. Nevertheless, these statistics and prognostic information differ among the different subgroups of medulloblastoma. In fact, a radical paradigm shift has been observed in medulloblastoma research since the 2016 WHO Classification of Tumors of the Central Nervous System, which divided medulloblastomas into four molecularly stratified subgroups: wingless (WNT) activated, sonic hedgehog (SHH) activated, group 3, and group 4 [4] (Table 1). This stratification was reiterated in the 2021 classification with further molecular-based subclassifications [5]. The unveiling of the vast differences in the genomes, epigenomes, transcriptomes, and proteomes of these subgroups ignited the search for subgroup-specific targeted therapies that can offer superior clinical outcomes with less devastating systemic side effects.

This paper aims to provide a summary of the neurodevelopmental and molecular profiles of medulloblastoma subtypes. Moreover, it builds on this characterization to further elaborate on the targeted therapeutic options that have been investigated in each subgroup, and it explores the promising potential of combination therapies as the future of medulloblastoma research and clinical practice.

## 2. Neurodevelopmental and Molecular Underpinnings of Medulloblastoma Subgroups

Several efforts of transcriptional profiling of medulloblastomas have demonstrated how these tumors closely recapitulate their physiological cellular counterparts in the developing cerebellum [9,10,11,12]. Even more importantly, the putative cells of origin of each medulloblastoma type seem to arise in specific spatiotemporal niches of the developing cerebellum, each giving rise to different cerebellar cellular lineages [9,10,11,12]. Therefore, the combination of timing, location of the initiating mutations, and the cell type affected by these mutations dictates the resulting medulloblastoma subgroup (Figure 1).

### 2.1. Cerebellar Embryonal Development

Given the strong developmental footprint of medulloblastomas, understanding the basic processes of cerebellar development is essential to uncover the embryonal origins of the different medulloblastoma subgroups. In the developing cerebellum, two main germinal epithelia can be identified: the ventricular zone (VZ), which is marked by the PTF1A marker and will generate the whole GABAergic cell lineage (including Purkinje cells) [13], and the rhombic lip (RL), the dorsal-most portion of the hindbrain proliferative neuroepithelium [14]. The RL is identified by the MATH1 marker and can be divided craniocaudally into the upper rhombic lip (URL) and the lower rhombic lip (LRL) [15].

On a purely morphological basis, this anatomical division relates to the segmentation of the hindbrain into rhombomeres (r1–r7) along the craniocaudal axis, with the URL developing from the dorsal pole or r1 and the LRL deriving from r2 to r7 segments [16]. On the other hand, from a gene expression standpoint, the URL is defined by the MATH1 marker gene as well as NEUROD1 [16]. In comparison, the LRL expresses PTF1a, WNT1 in its dorsoventral portion, and MATH1 in its dorsal portion [17], and it will partake in generating mostly extracerebellar neurons [18] including those forming the cochlear and pre-cerebellar nuclei [17]. The MATH1-expressing URL and dorsal LRL generate the cells of the glutamatergic lineage, including cerebellar nuclei neurons, granule cell progenitors (GCPs), and unipolar brush cells (UBCs) [13,17,19]. Overall, all cerebellar neurons are generated by both the PTF1a+ VZ and the MATH1+ portions of the RL [20].

Starting from post-conception week 10, the RL splits in two substructures: the RL ventricular zone (RLvz) and the RL subventricular zone (RLsvz). They are divided by a vascularized bed which is evident by post conception week 11 in humans [21]. The RLvz is characterized by Ki67-rich Sox2+ cells, while the RLsvz has Ki67-rich Sox2-sparse cells [21]. Additional markers include Wntless (WLS), CRYAB, SOX2, and PAX6 for the RLvz and TBR2 and EOMES for the RLsvz, while a common marker for all RL cells is LMX1A [21].

### 2.2. WNT-Activated Medulloblastoma

The WNT-activated subgroup constitutes approximately 10% of all medulloblastoma cases [2]. The median age at diagnosis is 11 years, with an almost equal male-to-female ratio [22]. This tumor group is believed to arise from the pontine mossy fiber precursor cells [9,23] of the extracerebellar LRL [9,10,12] that harbor somatic mutations in β-catenin (encoded by the gene *CTNNB1*), DDX3X, and SMARCA4 or germline APC mutations responsible for constitutive WNT signaling [22,24,25] (Figure 1). Using a Similarity Network Fusion approach on 763 primary medulloblastoma samples, Cavalli et al. identified two subtypes of WNT-activated medulloblastoma: WNT-α, typical of younger patients and characterized by monosomy at chromosome 6, and WNT-β, characteristic of adult patients and devoid of monosomy 6 [26]. The latter group featured a worse prognosis compared to the pediatric group.

Hovestadt et al. [9] conducted a single-cell transcriptomics analysis of medulloblastoma to highlight the peculiar cellular states of malignant cells in each subgroup. They found that WNT-amplified medulloblastoma cells exist in a differentiated neuronal-like state and display four transcriptional metaprograms with distinctive cellular signatures: WNT-A, associated with cell cycle activity; WNT-B, related to protein biosynthesis and metabolism; WNT-C, mirroring neuronal differentiation; and WNT-D, featuring the expression of early response and WNT pathway genes. Further scoring of transcriptional metaprograms outlined a developmental hierarchy within the WNT subgroup of medulloblastoma, in which tumor cells with high WNT-B and low WNT-C/WNT-D signatures possess proliferative capacity and drive tumor growth.

From the epigenetic perspective, the WNT subgroup of medulloblastoma features mutations in the epigenetic regulators of a sparse set of genes. Most of these mutations are shared with other subgroups, including those in *ARID1A*, *ARID2*, *CREBBP*, *MLL2/KMT2D*, and *SMARCA4*, a gene belonging to the SWI/SNF family of ATP-dependent chromatin remodeling complexes [8,27,28]. Promoter methylation of the tumor suppressor CDH1 is instead restricted to this MB type, a finding supported by studies showing the importance of CDH1 in regulating the WNT signaling in the LRL [8,27,29].

### 2.3. SHH-Activated Medulloblastoma

This subgroup is the most common group of medulloblastoma in kids less than three years of age and in adults (>18 years of age), with an approximately 2:1 male-to-female ratio [14,15]. It is thought to arise from GCPs and granule cells in the URL, presumably from the portion giving rise to the external granular layer [10,12,22]. The clinical outcome in this type is heterogeneous and strongly dictated by the underlying transcriptional and cellular activity [30]. Cavalli et al. [26] identified four main subtypes of SHH-activated medulloblastoma, along with their main mutations and age groups: SHH-α, typical of 3–16-year-old patients, has the worst prognosis among this group and harbors TP53 mutations in one-third of the cases; SHH-β and SHH-γ groups are more prevalent in infants (1.3 and 1.9 years of age, respectively), with SHH-β displaying a poorer prognosis compared to SHH-γ due to increased rate of metastatic dissemination. Finally, SHH-δ group occurs mainly in adults, features TERT promoter mutations, and has an overall good prognosis.

Hovestadt et al. [9] demonstrated three transcriptional metaprograms in SHH-amplified medulloblastoma cells. These metaprograms are related to cell cycle activity, undifferentiated progenitors, and neuronal differentiation (SHH-A, SHH-B, and SHH-C, respectively). In accordance with epidemiologic findings, this work also showed that the cellular origin of SHH medulloblastoma is dichotomic according to age: pediatric tumors feature cells in granule neuron-like states expressing high levels of NEUROD1, while adult tumors express higher levels of MATH1 and feature cells in either granule neuron progenitor-like state or in a mixed state between UBCs and granule neurons (Figure 1). These genes are prototypical markers of the URL during cerebellar development [16].

Adult SHH-activated medulloblastomas display a higher mutation burden compared to their pediatric counterparts, particularly with mutations associated with the SHH pathway, mainly Patched1 (PTCH1) and Smoothened (SMO), as well as mutations in CREB binding protein (CREBBP), BRPF1, and TERT promoter [22]. Additionally, adult tumors display higher proportions of cells in the undifferentiated granule neuron progenitor-like state compared to pediatric tumors [31]. Overall, these findings may help explain the differences in therapeutic outcomes in SHH-activated medulloblastomas and their susceptibility to targeted therapies. Moreover, Hovestadt et al. [9] noted that the SHH-B metaprogram was the only proliferating compartment associated with the expression of SHH pathway genes, but they did not explore which age groups had the highest score for SHH-B and how this expression affects response to therapy. Future studies should aim to characterize the effects of targeted therapy on the different transcriptional metaprograms of SHH-activated medulloblastoma and stratify those results according to age.

Recurrent epigenetic alterations in SHH-activated medulloblastoma have been described in MLL2/KMT2D and MLL3/KMT2C, two lysine methyltransferases associated with an active chromatin state and the H3K4me2/3 status [8,32,33]. These mutations have also been reported for group 3 and group 4 tumors. Alternatively, subtype-specific mutations have been reported in NCOR2 and LDB1, two chromatin remodelers belonging to the nuclear co-repressor (N-CoR) complex [33]. Importantly, N-CoR dysregulation has been described as a crucial driver for SHH medulloblastoma onset [33].

### 2.4. Group 3 Medulloblastoma

Group 3 medulloblastomas bear the worst prognosis among all subtypes. Approximately 50% of group 3 tumors feature dissemination along the neuroaxis at diagnosis [34]. Cavalli et al. [26] classified group 3 tumors into three discrete subcategories: Group 3α, present in infants with metastatic dissemination but associated with better outcomes; Group 3β, occurring in an older age group and having a reduced metastatic rate; and Group 3γ, which has the worst prognosis among all subgroups. This last subgroup has been identified as originating from the earlier RLvz [10].

Luo et al. demonstrated that group 3 medulloblastoma cells resemble transitional cerebellar progenitor (TCP) cells, a transient-amplifying proliferating compartment physiologically more present in the RLvz, RL transitional zone (RLtz), and to a lesser extent in the RLsvz [11] during neurodevelopment. TCPs may be the cells of origin of group 3 medulloblastoma and can be identified by two signature markers, HNRNPH1 and SOX11. At the molecular level, the juxtaposition of HNRNPH1 and SOX11 super-enhancers to MYC cis-regulatory elements, through a mechanism of distance looping, appears to be responsible for MYC overexpression and the abnormal proliferation of group 3 tumor cells [11]. The proportion of TCP-like tumor cells in group tumors correlated with the rate of dissemination to the spinal cord and leptomeninges [11]. Therefore, sampling HNRNPH1 and SOX11 in tumor specimens and correlating their expression with the risk of metastasis may represent a future prognostic strategy in this group of medulloblastoma.

The study by Northcott et al. [25] employed the Cis Expression Structural Alteration Mapping (CESAM) technique to demonstrate that enhancer hijacking of growth factor independent 1 (GFI1) or GFI1B proto-oncogenes potentiate the effects of MYC amplification and further promotes tumor proliferation in group 3 medulloblastomas. The action of GFI1 is also mediated by LSD1, a histone lysine demethylase that is a potential treatment target for both group 3 and group 4 tumors [22].

Hovestadt et al. [9] demonstrated that prototypic group 3 tumors displayed cells in an undifferentiated progenitor-like metaprogram, characterized by ribosomal and translational initiation/elongation and MYC target gene expression. They also showed that group 3 tumor cells appear stalled in an undifferentiated neural progenitor cell state, hinting to mutations inducing a block of neural differentiation. In support of this evidence, another study [35] showed that OTX2 amplification reduces the expression of downstream regulators of neuronal differentiation including PAX3 and PAX6, serving as a differentiation blocker for group 3 medulloblastoma cells and inducing downstream mammalian target of rapamycin 1 (mTOR1) activation for protein synthesis and translation/elongation factor genes, consistent with the findings by Hovestadt et al. [9].

It appears that multiple structural variants confer selective growth advantages to group 3 medulloblastoma cells. Amplification of OTX2 halts the process of differentiation in the undifferentiated progenitor cell compartment present in the early RL [35]. MYC oncogene amplification and upregulation of pathways involved in protein synthesis further promote cellular proliferation [11]. Finally, GFI1/GFI1B activation by enhancer hijacking increases the action of MYC [25]. Driver mutations have been shown to differ according to the DNA methylation subtype of each group 3 tumor [22,26] and may help explain the intratumoral heterogeneity of this group of medulloblastomas.

Finally, group 3 medulloblastomas are characterized by an array of epigenetic dysregulations, some of which include distance looping and enhancer hijacking to block differentiation and boost tumor cell proliferation. At the histone level instead, group 3 tumors harbor mutations in genes belonging to the lysine demethylase family (KDM), sharing several of these mutations with group 4 medulloblastomas [8]. Bromodomain (BRD) and extra C-terminal (BET)-containing proteins bind acetylated histones and recruit the transcriptional machinery to control MYC levels, which is crucial in group 3 onset [36,37,38].

Peculiar alterations in histone regulators can help explain the biology of group 3 tumor cells. In fact, group 3 medulloblastomas display mutations in the polycomb repressor complex 2 (PRC2) gene set, which is a crucial regulator of differentiation, proliferation, and cell identity [8]. Among the components of PRC2 lies the enhancer of zeste homolog 2 (EZH2), the catalytic partner of PRC2 that causes the addition of methyl groups to histone 3 to promote the H3K27me3 status, with consequent chromatin compaction and transcriptional repression [39]. EZH2 overexpression in group 3 medulloblastoma increases H3K27me3 and impairs H3K4 methylation, thereby keeping cells in a stem-like/progenitor state [40]. This finding may promote the maintenance of group 3 cells in an undifferentiated state and further boost their malignant potential.

### 2.5. Group 4 Medulloblastoma

Group 4 medulloblastomas are the overall most common type [2], spanning across all age groups and having a 2:1 male-to-female ratio [22]. They are believed to originate from cells of the glutamatergic lineage, particularly from UBCs and glutamatergic cerebellar nuclei (Glu-CN) neurons arising in the URL [9,41,42]. In the developing cerebellum, these cells are marked by glutamatergic and RLsvz-specific transcription factors including EOMES, LMX1A, and TBR2 [9,10,21]. Cavalli et al. [26] identified three main subgroups of group 4 medulloblastoma: group 4α, featuring MYCN amplification; group 4β, characterized by SNCAIP duplication; and group 4γ, displaying cyclin-dependent kinase 6 (CDK6) amplification.

In the analysis by Hovestadt et al., prototypic group 4 tumors expressed a differentiated neuronal-like metaprogram (Group 3/4-C), including genes associated with the neuronal lineage [9]. Hendrikse et al. [10] showed that mutations in the CBFA gene complex (most notably alterations in KDM6A and enhancer hijacking of PRDM6 and GFI1/GFI1B) in UBCs, the last cells to develop from the RLsvz, are responsible for group 4 medulloblastoma development. Herein, GFI1 and GFI1B oncogenes are abnormally expressed in both group 3 and group 4 tumors in a mutually exclusive fashion [25]. In fact, local enhancer hijacking of GFI1 and distal enhancer hijacking of GFI1B drive medulloblastoma growth, either by cooperating with MYC to drive group 3 tumors or with other drivers to promote group 4 medulloblastoma development [43].

The PRDM6 gene encodes for a transcriptional repressor that uses histone 4 lysine 20 (H4K20) methyltransferase to induce gene silencing [22]. It is the most frequent somatically altered gene in group 4, being present in 17% of patients with this subtype of medulloblastoma [22] and featuring a more than 20-fold upregulation in group 4 tumors [25]. Moreover, it is located 600 kb downstream of the SNCAIP locus, a hotspot for tandem duplications that are unique to group 4 medulloblastoma [44]. By utilizing the CESAM technique, Northcott et al. [25] revealed the presence of structural variants bringing the SNCAIP super-enhancers closer to PRDM6, inducing its activation and overexpression.

Like group 3 tumors, group 4 medulloblastomas also display epigenetic alterations at multiple levels. In fact, both enhancer hijacking and histone alteration mechanisms are found in this subtype. From the histone mutation standpoint, group 4 tumors feature a prototypical inactivation of KDM6A/UTX, a member of the lysine demethylase family [27,36]. The interesting point lies in the fact that KDM6A/UTX mutations, which are more common in this group, have the opposite effect of EZH2 amplifications typical of group 3 tumors. In particular, while the former promotes transcription by removing methyl groups and acetylating histones, the latter causes histone demethylation and chromatin compaction. Further, these two mutations are mutually exclusive in group 3 and group 4 medulloblastomas [40]. This finding may help shed light on the differences between these subtypes.

### 2.6. Intermediate Group 3/Group 4 Medulloblastoma

There is an ongoing debate about the cellular and transcriptional nature of intermediate group 3/group 4 tumors. Luo et al. [11] have reported the presence of distinct group 3 and group 4 subpopulations in intermediate tumors by single-cell clustering. This finding is in direct contrast with those of Williamson et al. [45], showing that group 3 and group 4 tumor cells exist along a common transcriptional continuum that reflects the glutamatergic lineage of cerebellar development. In this last model, group 3 cells resemble more primitive cells in the rhombic lip while group 4 cells are closer to the more differentiated excitatory UBC cohort. Moreover, the DNA methylation subtypes of group 3/4 tumor cells also lie along this same continuum. The distribution of single-medulloblastoma tumors along this spectrum is influenced by both transcriptional status and methylation subtype and appears to have prognostic significance, particularly in the first five years post-diagnosis.

In the analysis by Hovestadt et al., intermediate group 3/group 4 tumors consisted of an admixture of both DNA methylation subtypes and metaprograms from the two extremes. The authors interpreted these findings as reflecting a cell state continuum rather than a combination of distinct cellular populations [9]. Similarly, a recent work by Smith et al. [41] identified a common developmental origin for both group 3 and group 4 medulloblastomas in the RLsvz. The whole spectrum of group 3, group 4, and intermediate group 3/group 4 tumors lies along the differentiation axis of cells arising from the RLsvz, with early cells that bear a photoreceptor gene signature developing into group 3 medulloblastomas and late cells with a UBC signature giving rise to group 4 medulloblastomas. In this context, intermediate group 3/4 tumors featured a mixed photoreceptor-like and UBC-like expression profile, and a specific gene signature still lining along the RL-UBC developmental axis (DDX31-GFI1B, OTX2, and MYCN) (Figure 1).

In this emerging perspective, group 3 and group 4 medulloblastomas may share a developmental origin in the RL, supporting the idea of a transcriptional and DNA methylation gradient encompassing group 3, group 4, and intermediate group 3/4 tumors. The proportion of differentiated cell states may reveal the precise biology of each individual tumor and determine its position along this axis. In fact, while group 3 tumors are comprised only up to 10% differentiated neuronal-like cells, group 4 tumors may almost entirely be composed of differentiated UBC-like and Glu-CN-like cells [31]. On the other hand, intermediate group 3/4 medulloblastomas feature a mixture of undifferentiated and mature neuron-like cells [9,31]. Finally, differences in epigenetic alterations, which were discussed previously, may provide another way to distinguish group 3 and group 4 tumors along the group 3/4 tumor spectrum.

## 3. Subgroup-Specific Targeted Therapies in Medulloblastoma

### 3.1. WNT-Activated Medulloblastoma

This subgroup of medulloblastomas is known for its excellent prognosis and high survival rates. Indeed, it has been reported by multiple studies that the 5-year survival rate of patients less than 16 years of age with WNT-activated medulloblastomas is greater than 90% following standard treatment with surgery, chemotherapy, and radiation therapy [34,46,47]. The favorable prognosis of this subtype is attributed to the greater penetration of chemotherapeutic agents due to the aberrant vasculature and disrupted blood–brain barrier function in the vicinity of these tumors [48]. In this context, when compared to other subtypes, WNT medulloblastomas were found to have more dense and tortuous vessels with fenestrated endothelial lining and disrupted tight junctions. This leaky phenotype is due to the suppression of the WNT pathway, which is crucial for proper angiogenesis, in the endothelial cells by paracrine signaling from neighboring WNT-activated tumors. In specific, these tumors secrete WNT inhibitors, such as WNT Inhibitor Factor 1 (WIF1) and Dickkopf 1 (DKK1), potentially as part of a negative feedback loop. These WNT inhibitors diffuse and execute their angiogenesis-disrupting effects on the nearby vasculature, thus producing the leaky phenotype [48].

Despite the excellent prognosis achieved with the current treatment regimen, the used modalities are not without risks and complications. One of the major concerns for using the high doses of craniospinal irradiation that are used in cases of medulloblastoma in the pediatric population is the association with long-term neurocognitive impairment [49]. Therefore, research on WNT-activated medulloblastomas has shifted towards de-escalation trails that aim at reducing the unnecessarily high doses of radiation and chemotherapy in this well-responding group. The importance of these dose de-escalation efforts is supported by evidence on the reduced intellectual burden in WNT medulloblastoma survivors who received lower radiation doses [50]. Investigating the possibility of de-escalated therapies in WNT-activated medulloblastoma gained great traction after the results of the Children’s Oncology Group (COG) trial, ACNS0331 (NCT00085735), were published. This trial showed that reducing the dose of craniospinal irradiation resulted in lower survival rates for patients with medulloblastoma; however, a subgroup analysis of patients with WNT-activated medulloblastomas showed favorable outcomes for dose de-escalation in this subgroup only [51]. Based on that, another trial by COG (NCT02724579) was initiated and is currently ongoing to assess the outcomes of reduced dose radiotherapy (18 Gy craniospinal irradiation and 36 Gy to the tumor bed) and reduced chemotherapy (eliminating vincristine during radiotherapy and using a reduced maintenance dose) in WNT-activated medulloblastoma (clinicaltrials.gov (accessed on 24 July 2023)). Similarly, the FOR-WNT2 clinical trial (NCT04474964) is also currently recruiting and investigates the impact of the same reduced dose of radiation therapy on clinical outcomes in this subgroup of medulloblastoma. However, it is worth mentioning that previous attempts to avoid radiation therapy altogether or use focal radiation therapy only instead of craniospinal irradiation were aborted due to the high relapse rates [52].

Although the de-escalation trials provide a promising route for attenuating the deleterious side effects of radiation and chemotherapy, new targeted agents are still needed to replace these traditional therapies or at least help in further reducing the needed doses. In this context, targeting the WNT pathway might present itself as a rational option in this subgroup (Figure 2); however, there are several challenges that arise when attempting to target this pathway. First, the WNT pathway plays a pivotal role in bone formation, hematopoiesis, tissue repair and regeneration, and homeostatic balance in several organs, and thus multiple deleterious effects and interruption of developmental processes can be anticipated if this pathway is disrupted [53]. Second, there is valid concern that targeting the WNT pathway might jeopardize the favorable features seen in WNT-activated medulloblastomas, such as their leaky vasculature and excellent response to chemotherapy. Finally, the complexity of this signaling pathway makes it difficult to determine which players in the cascade are the ideal targets for pharmacotherapeutic approaches [53].

Despite these challenges, there are several molecular targets that are worth being explored in this subtype. Mutant DDX3, an RNA helicase, has been shown to augment the activity mutant β-catenin, and the two molecules synergistically increased the proliferation of medulloblastoma cell lines [54]. Interestingly, using RK-33, a small-molecule inhibitor of DDX3, resulted in inhibition of the WNT pathway and G1 arrest in the medulloblastoma cells in vitro [55]. Not only that, but RK-33 was also associated with increased radiosensitivity of in vitro DAOY and UW228 cell cultures and of DAOY flank tumors in nude mice [55]. Another molecule that has been investigated as a potential target for interfering with the WNT pathway is tankyrase (TNKS), which is implicated in the regulation of this pathway. TNKS inhibitors induce the accumulation of Axin, thus further stabilizing the complex (Axin, APC, GSK3β, and CKIα) that tags β-catenin for destruction [56] (Figure 2). Herein, XAV-939, one of the earliest TKNS inhibitors, was shown to inhibit WNT signaling in DAOY and ONS-76 cell lines. Furthermore, XAV-939 treatment disrupts the DNA repair abilities of these cell lines and increases their sensitivity to ionizing radiation [57]. However, it is important to mention that XAV-939 can also inhibit the poly-ADP-ribose polymerase 1 (PARP1). So, it is not clear whether the XAV-939-induced radiosensitivity is mainly due to its TNKS or PARP1 inhibitory effects. Nevertheless, TNKS inhibitors are worth being considered as promising agents in WNT-activated medulloblastomas especially with the advent of newer and more effective agents in this class [58]. Additionally, fenretinide, which is a synthetic analogue of all-trans retinoic acid, has shown that it possesses anti-WNT properties. In specific, the expression of WNT3A and its downstream effectors was reduced in DAOY and ONS-76 cells after treatment with fenretinide [59]. Moreover, fenretinide was able to inhibit the proliferation of these cell lines in vitro [59]. However, further studies are warranted to confirm whether fenretinide’s anti-cancer effects are reproducible in animal models and whether they are solely due to WNT inhibition or to its other effects on cellular oxidative balance [60].

On top of the aforementioned approaches, there are multiple tractable targets that can be exploited to hinder WNT signaling and that have shown promising results in other types of cancer with WNT upregulation. However, these targets have not been well-studied in the context of medulloblastoma yet. For instance, the interaction between CREBBP and *CTNNB1* is crucial for the transcriptional activation of this gene, and thus for WNT signaling (Figure 2). Here, it is worth mentioning that PRI-724, which inhibits the CREBBP: *CTNNB1* interaction, has shown promising results as a combination therapy in a phase 1 trial against advanced pancreatic adenocarcinoma (NCT01764477) [61]. In addition to inhibiting the WNT pathway itself, other pathways that are commonly overactivated in the WNT subgroup of medulloblastomas can serve as potential targets. In specific, the ALK pathway has been proven to be a commonly overexpressed pathway in WNT-activated medulloblastomas and was even suggested as a novel biomarker for the diagnosis of this subgroup [62,63]. Therefore, ALK inhibitors might be appealing pharmacologic agents that deserve to be investigated in these tumors.

Furthermore, the emergence of epigenetic profiling and targeting techniques created novel routes for molecular-based therapeutics in medulloblastoma. Herein, the role of histone deacetylation was explored in these tumors, and it was found to contribute to the downregulation of the WNT inhibitor DKK1 in medulloblastoma. In fact, the use of the histone deacetylase inhibitor trichostatin A resulted in rescuing the expression of DKK1 and a subsequent increase in the apoptotic cell death of medulloblastoma cells [64]. Such epigenetic interventions are worth being further studied to confirm their applicability and efficacy.

### 3.2. SHH-Activated Medulloblastoma

The SHH-activated subgroup has the most adequately characterized molecular and genetic profile offering a wide array of appealing targets. Subsequently, a myriad of therapeutic agents has been investigated to modulate the SHH pathway or other oncogenic pathways that interact with it (Figure 3). In a nutshell, the SHH signaling cascade is initiated with the binding of the SHH ligand to the PTCH transmembrane receptor, thus removing the blockade of the latter on SMO, which is a G protein-coupled receptor. In its turn, SMO translocates to the primary cilium where it causes the activation of proteins belonging to the glioma-associated oncogene (GLI) family by triggering their dissociation from their repressor SUFU. Upon that, GLI proteins translocate to the nucleus and orchestrate the transcription of effector genes involved in the actions of the SHH pathway. In addition to this canonical pathway, multiple alternative non-canonical pathways have been shown to activate SHH signaling downstream of SMO [65].

The road towards developing targeted therapies that can disrupt SHH signaling started with SMO inhibitors. In specific, SMO is composed of two extracellular domains called the cysteine-rich domain and the linker domain, a transmembrane domain consisting of seven membrane-spanning subunits, and an intracellular domain responsible for downstream signaling. Herein, cyclopamine is the earliest SMO inhibitor investigated in the context of SHH-activated medulloblastoma, and it acts by binding to the transmembrane portion of the receptor (Figure 3). However, several concerns regarding the safety of this drug have arisen and led to the abortion of further clinical investigations regarding its utility in medulloblastoma [66]. Nevertheless, cyclopamine ignited the search for other small molecules that can suppress the activity of SMO with comparable efficacy and more acceptable safety profiles. The most popular among these are vismodegib (GDC-0449) and sonidegib (LDE-225), which have shown promising effects in preclinical models [67,68] and made it to clinical trials. Interestingly, both drugs have already been FDA-approved for use in locally advanced basal cell carcinoma [69]. In the context of medulloblastoma, the Pediatric Brain Tumor Consortium (PBTC) conducted a phase I clinical trial (PBTC-025, NCT00822458) that confirmed the tolerability of vismodegib [70], and then followed this trial with two phase II trials involving adult patients (PBTC-025B, NCT00939484) and pediatric patients (PBTC-032, NCT01239316) with recurrent or refractory medulloblastomas. The results of these phase II trials showed an increased progression-free survival in adult patients with SHH medulloblastoma compared to those with non-SHH medulloblastoma, suggesting an effective role for vismodegib in the former subgroup [71]. Currently, an ongoing phase II clinical trial (NCT01878617) by St. Jude’s Children’s Research Hospital that assigns different interventions based on molecular subgroup and risk stratification investigates the efficacy of vismodegib in skeletally mature patients belonging to both standard-risk and high-risk SHH subgroups. In a similar fashion, sonidegib has shown good safety and efficacy in pediatric and adult patients with progressive or refractory SHH-activated medulloblastomas during a phase I/II trial (NCT01125800) [72]. In addition, an actively recruiting randomized controlled phase II trial (PersoMed-I, NCT04402073) by the European Organisation for Research and Treatment of Cancer aims to assess the efficacy of sonidegib with reduced-dose radiotherapy in post-pubertal patients with SHH-activated medulloblastomas. Although vismodegib and sonidegib are the most popular agents that target the transmembrane domain of SMO, there are several other drugs that have a similar mechanism and that have also shown promising results in animal models of medulloblastoma, such as MK-4101, L-4, and nilotinib; however, these agents have not entered clinical trials yet [66]. Additionally, the agents ALLO1 and ALLO2 have been found to inhibit SMO through a different mechanism involving the cysteine-rich domain and have shown anti-proliferative effects in medulloblastoma cells [66] (Figure 3).

Although the results of the mentioned trials involving vismodegib and sonidegib were encouraging in the SHH-activated subgroups, the development of resistance to these agents was reported clinically. One identified mutation that causes this resistance is the D473H mutation of the SMO protein. Herein, a relatively newer agent, taladegib (ENV-101, LY2940680), has been shown to overcome this resistance method and suppress SMO in its wild-type and mutated forms [73]. Taladegib is currently being investigated in a phase II clinical trial (NCT05199584) involving patients with solid tumors and PTCH1 loss of function mutations, which is a common mutation in SHH-activated medulloblastomas. Recently, Ji et al. synthesized a taladegib-based compound that elicits a more potent inhibition of SMO and a more significant attenuation of DAOY cells proliferation [74]. On the other hand, the antifungal itraconazole inhibits the activity of SMO through halting its translocation to the cilium (Figure 3), and it appears to be effective against D477G-mutant medulloblastoma mouse models that are resistant to other SMO antagonists [75,76]. In a similar fashion, the repurposing of the antiparasitic drug mebendazole has gained wide attention for its promising potential as an anti-cancer drug [77]. In specific, mebendazole was also found to inhibit SHH signaling by hindering the formation of the primary cilium, and thus resulted in decreased proliferation of DAOY cells in vitro and extended the survival of SHH medulloblastoma orthotopic models [78,79]. Importantly, the inhibitory effect of mebendazole was also present when used on vismodegib-resistant models. A phase 1 clinical trial of mebendazole in progressive/refractory pediatric brain tumors, including medulloblastoma, was conducted at the Sidney Kimmel Comprehensive Cancer Center at Johns Hopkins (NCT02644291), but the results are yet to be published.

In addition to resistance at the level of SMO itself, other genetic alterations were commonly found in non-responders to vismodegib and sonidegib. As expected, these genetic alterations consist of mutations involving members of the SHH pathway downstream of SMO, such as SUFU and GLI1 [71,80,81]. Not only that, but also the crosstalk between SHH and other oncogenic pathways that are commonly overactivated in medulloblastoma, such as PI3K/mTOR and RAS/REF/MEK, can play a significant role in evading SMO inhibition [6].

The aforementioned challenges have geared the research efforts towards targeting the SHH pathway at points that are further downstream of SMO. In specific, disrupting the production and action of GLI proteins at different levels has gained significant attention (Figure 3). For instance, BET proteins have been recognized as tractable epigenetic targets in several cancer types, including medulloblastoma. In specific, BRD4 interacts with the promoter regions of GLI1 and GLI2 and enhances their transcription. Notably, JQ1, a BRD4 inhibitor, was shown to inhibit the SHH-mediated proliferation of several tumors, including medulloblastomas, and to overcome their resistance to SMO antagonists [82]. A phase I clinical trial (NCT03936465) is currently recruiting pediatric patients with solid tumors or lymphoma, with a separate arm for refractory or metastatic CNS tumors, to assess the safety of the BRD inhibitors BMS-986378 and BMS-986158. Another avenue for modulating the transcription of GLI proteins is targeting the casein kinases alpha 1 and 2 (CKα1 and CK2). CKα1 is a negative regulator of GLI transcription factors, while CK2 is a positive one. Expectedly, both CKα1 agonists (pyrvinium and SSTC3) and CK2 antagonists (CX-4945) have shown significant efficacy in SHH-activated medulloblastoma mouse models, even in the presence of the TP53 mutation which, as previously mentioned, imparts a worse prognosis [66]. In fact, an actively recruiting clinical trial (NCT03904862) investigates the safety and tolerability of CX-4945 in skeletally immature patients with refractory/recurrent SHH-activated medulloblastoma (phase I) and its efficacy in skeletally mature patients with refractory/recurrent SHH-activated medulloblastoma (phase II). Moreover, the CDK7 is implicated in the transcriptional regulation of GLI and has been investigated as a potential target to disrupt SHH signaling. The CDK7 inhibitor, TZH1, has shown significant potency in suppressing the SHH-mediated proliferation of medulloblastoma cells, including those that are resistant to SMO inhibitors [83]. In addition to targeting GLI proteins at the transcriptional level, direct inhibitors of the protein have been discovered and evaluated (Figure 3). In this context, the GLI antagonist 61 (GANT61) and arsenic trioxide (ATO) have both shown promising results as direct inhibitors of GLI proteins in medulloblastoma. Specifically, GANT61 was able to attenuate the proliferation and migration of DAOY cells, induce their apoptosis, augment their response to cisplatin, and sensitize them to particle radiation (protons and carbon ions) [84,85]. ATO inhibited the proliferation of SHH-activated medulloblastoma cell lines both in vitro and in vivo [86,87]. Moreover, it increased the sensitivity of TP53-mutated SHH-activated medulloblastoma cells to radiation [87]. It is worth mentioning that ATO has already entered phase I and phase II clinical trials for other pediatric brain tumors and has shown encouraging results [88,89].

Apart from targeting the SHH pathway itself at different levels, several groups have investigated the utility of targeting other pathways or effectors that are commonly overactivated in this subgroup of medulloblastomas. For instance, the Mesenchymal–Epithelial Transition factor (cMET) was found to be upregulated in SHH-activated medulloblastoma and to correlate with worse prognosis [90]. Hence, the cMET inhibitor foretinib was investigated for the treatment of this subgroup of medulloblastoma, and it showed acceptable penetration of the blood–brain barrier and significant suppression of the proliferation and migration of SHH-activated medulloblastomas in xenograft mouse models [90]. Another molecule that has been found to possess interesting interactions with the SHH pathway is the AMP-activated protein kinase (AMPK). In this context, AMPK is a cellular sensor of a low energy state that regulates energy-demanding function and shuts them down when needed, and among these functions is the activation of the SHH pathway. Indeed, activated AMPK has been proven to negatively regulate GLI1 in a direct manner by triple phosphorylating it, undermining its stability, and promoting its degradation [91]. In addition, AMPK has been suggested to regulate GLI1 in an indirect manner also, mainly through its suppression of the activity of the mTOR/S6K pathway, which upregulates GLI1 expression [91]. Hence, these inhibitory effects that AMPK exerts on the SHH pathway downstream of SMO are worth being explored for their potential therapeutic benefits. In fact, the antidiabetic drug metformin has been investigated as an anti-cancer agent due to its ability to promote the activity of AMPK, suppress the action of mTOR, and subsequently attenuate SHH/GLI signaling. Promising effects of the drug have been documented in the context of prostate cancer, gastric cancer, and hepatocellular carcinoma [92,93,94], and it is, therefore, worth being further investigated in the setting of SHH-activated medulloblastoma. On another note, with the increased interest in epigenetics and their utility in cancer therapeutics, epigenetic profiling of SHH-driven medulloblastomas exhibited an increased expression of the miR17~92 polycitron in this subgroup and a synergistic effect between this overexpression and the SHH pathway to augment the growth of cancer cells [95]. Herein, the use of locked nucleic acid (LNA) antisense oligonucleotides (anti-miRs) to inhibit miRNAs has been explored. Specifically, anti-miR17 and anti-miR19, which inhibit miRNAs 17 and 19-a that belong to the miR17~92 complex, were able to suppress the proliferation of SHH-activated medulloblastoma cells in vitro and hinder the progression of tumors belonging to this subgroup in vivo [96]. Not only that, but differences in the immune profiles of medulloblastoma subgroups have been exploited to derive subgroup-specific immunomodulators. In this context, Tumor-associated macrophages (TAMs) were noticed to play a cancer-promoting role in medulloblastomas belonging to the SHH subgroup with SMO mutations [97]. Hence, the treatment of mice harboring SMO-mutated SHH-activated medulloblastomas with PLX5622, an inhibitor of the colony-stimulating factor 1 receptor (CSF1R), resulted in a reduced proportion of TAMs in the tumor microenvironment, shrinkage in tumor sizes, and prolonged survival of the mice [97].

A field that is attracting significant interest is the identification and targeting of cancer stem cells in brain tumors. This is because these cells are suggested to be major contributors to recurrence and resistance to conventional therapies [98]. Therefore, the identification of biomarkers that are characteristic of cancer stem cells in medulloblastoma has been crucial for deriving modalities to target these cells more effectively. For instance, CD15-positive cells were identified as tumor progenitor cells in an SHH medulloblastoma mouse model, and a large proportion of these cells was found to be in the G2/M phase of the cell cycle. Hence, targeting players that are specific to this phase was hypothesized as an effective method to abolish the proliferative potential of these cells. Indeed, inhibiting polo-like kinase (PLK) and Aurora kinase (AURK), which are pivotal for the G2/M transition of the cell cycle, using BI-2536 and VX-680 (tozasertib), respectively, resulted in increased apoptosis of SHH-driven medulloblastoma cells in vitro and in vivo [99]. Additionally, single-cell transcriptomics revealed that the OLIG2/SOX2 axis is especially overactivated in actively cycling progenitor cells in SHH-driven medulloblastomas and is a significant contributor to their self-renewal capacity. In fact, OLIG2 expression was also observed to be prominent in recurrent tumors, suggesting a role for this molecule in resistance to treatment [100]. Since OLIG2 is a nuclear transcriptional factor that is difficult to target directly, Zhang et al. identified other targetable effectors that mediate the actions of OLIG2 in these progenitor cells. Herein, the HIPPO-YAP/TAZ and the AURK/MYCN pathways are major mediators of OLIG2 actions. Notably, the combination of CD532, which disrupts the interaction between AURK and MYCN, and verteporfin, a YAP inhibitor, was associated with a dramatic suppression of tumor growth both in vitro and in vivo and an increased survival of SHH-activated medulloblastoma-bearing mice [100].

At any rate, with the discovery of novel agents that can disrupt the SHH pathway at multiple layers or target other oncogenic pathways that synergize with it, the potential for combination therapies that can achieve more efficacious results and reduce resistance to treatment is reinforced. For instance, in order to avoid the non-canonical activation of the SHH pathway by the oncogenic PI3k pathway, the concomitant use of PI3K inhibitors with sonidegib was attempted. Indeed, this combination impeded the development of resistance in SHH-driven medulloblastoma cells [101]. Similarly, the combination of an SMO inhibitor with the PLK inhibitor BI-2536 exhibited superior responses as compared to the use of the former agent alone [99]. Not only that, but evidence suggests that combination therapies can also be utilized against populations of tumor progenitor cells in medulloblastoma. In specific, the chemokine receptor CXCR4 was found to be frequently expressed with CD15, indicating a role for the former in progenitor cells [102]. Intriguingly, the co-inhibition of CXCR4 and SMO using AMD3100 and vismodegib, respectively, attenuated the proliferation of SHH-driven medulloblastoma flank and intracerebellar xenografts [102]. Finally, it is worth noting that an active phase I trial by the St. Jude Children’s research hospital (NCT03434262, SJDAWN) compares multiple combinations of molecularly driven therapies in pediatric patients with medulloblastoma, and it includes an arm that evaluates the combination of sonidegib with ribociclib, a CDK4/6 inhibitor, in refractory/recurrent cases of SHH-activated medulloblastoma.

### 3.3. Group 3 Medulloblastoma

This subtype of medulloblastomas has the greatest potential for invasion and metastasis and carries the most dismal prognosis. Consequently, it shows minimal response to conventional therapies. Hence, novel targeted therapies that are effective against this subgroup are greatly needed and have been a topic of interest for several laboratories worldwide. One of the factors that have been associated with a worse prognosis in this subgroup is the presence of MYC amplification. In this context, a relationship was discovered between BRD inhibitors and MYC activity. Specifically, the BRD4 inhibitor, JQ1, was shown to reduce the proliferation of group 3 medulloblastoma cells with MYC amplification and to prolong the survival of mouse models with tumors having these characteristics [103,104] (Figure 4). In a similar fashion, a novel in silico drug screening method, named DiSCoVER, predicted a potential role for CDK inhibitors in MYC-activated group 3 medulloblastomas [105]. Indeed, the CDK4/6 inhibitor, palbociclib, was proven to elicit anti-proliferative and pro-apoptotic effects against in vivo and in vitro models of this subgroup [105,106]. Likewise, the CDK1/5 inhibitor, alsterpaullone, has also shown significant inhibition of the MYC-dependent proliferation of against group 3 medulloblastoma cells [107]. Notably, the safety of palbociclib in pediatric patients has been confirmed in a phase I clinical trial (NCT02255461) involving pediatric patients with progressive or refractory brain tumors [108]. Further, a phase II trial (SJDAWN, NCT03434262) is currently active and evaluates the efficacy of ribociclib, another CDK4/6 inhibitor, combined with gemcitabine in recurrent/refractory group 3/4 medulloblastoma. In a surveillance of the phosphor-proteomic footprint of medulloblastoma, the DNA-dependent protein kinase PRKDC was predicted to play an important role in group 3 medulloblastomas [109]. Indeed, it was found that the levels of this kinase were elevated in this subgroup, and other experiments have uncovered its contribution to MYC stability. Additionally, PRKDC is known to play a role in non-homologous end joining during repair of DNA damage. Consequently, PRKDC inhibitors were investigated for their potential role in therapy, and they were found to elicit radiosensitizing effects in the D458 cell line, which resembles group 3 medulloblastoma with MYC amplification. However, PRKDC inhibitors did not show cytotoxic effects on their own [109]. In a parallel fashion, a large genomic analysis of MYC-driven group 3 medulloblastoma revealed an elevated expression of the inhibitory GABA-A receptor subunit alpha 5 (α5-GABAA). Therefore, an increased susceptibility of these cells to GABA agonists was hypothesized. Indeed, the α5-GABAA-specific agonist, QHii066, was able to induce apoptosis and cell cycle arrest of MYC-amplified medulloblastoma cells and to sensitize mice harboring D425 tumors, which mimic group 3 medulloblastoma with MYC amplification, to radiation therapy and cisplatin [110]. The anti-proliferative effects of this agonism were dependent on the activation of HOX5 and its related genes, which has already been implicated as an anti-cancer player in several cancers [110]. Based on these encouraging results, Jonas et al. tested the efficacy of several GABA agonists delivered to a model of intracranial group 3 medulloblastoma with MYC amplification, and they concluded that the benzodiazepine derivative KRM-II-08 is more potent that other agonists, including QHii066, and can offer a greater pro-apoptotic effect in these tumors [111]. Furthermore, targeting PLK1, SETD8, and facilitator of chromatin transcription (FACT) with their respective inhibitors onvasertib, UNC0379, and CBL0137 has also shown preclinical activity against MYC-amplified medulloblastoma [112,113,114]. It is also worth mentioning that the FDA-approved antiviral ribavirin, whose repurposing is currently being investigated in clinical trials for several cancer types [115], elicited anti-proliferative effects in D425 cells and showed a survival advantage in mice with intracranially implanted D425 tumors [116]. These effects of ribavirin were attributed to its ability to inhibit eukaryotic initiation factor 4E (eIF4E) and EZH2, both of which have been associated with MYC overexpression and group 3 medulloblastoma tumorigenesis [116]. Another FDA-approved drug that has shown good potential for repurposing is disulfiram, known as Antabuse. This drug, when combined with copper gluconate, induced apoptosis of group 3 and SHH-activated cell lines in vivo and prolonged the survival of mice with implanted cells from these lines [117]. Specifically, the effects of this combination were found to be mediated by the accumulation of nuclear protein localization protein-4 (NPL4) and the subsequent induction of cell death. Moreover, disulfiram plus copper treatment was shown to suppress DNA repair mechanisms, thus contributing to its lethal effect and suggesting a potential role as a sensitizer to radiotherapy or certain chemotherapies [117].

Based on the aforementioned therapeutic approaches, one can safely say that targeting MYC-amplified group 3 medulloblastoma has attracted a great majority of the research in this subtype. This overrepresentation of MYC-amplified group 3 medulloblastomas in the literature is, in fact, a reflection of the overabundance of cell lines that mimic this specific molecular profile [26]. Although MYC-activation is one of the hallmarks of this subgroup and inflicts a worse prognosis, it is only present in approximately 17% of group 3 tumors. Hence, several groups explored the utility of targeting other overactivated pathways that this subgroup might be dependent on. In particular, GFI1/GFI1B overexpression was found to be a present in 15–20% of group 3 medulloblastoma and to play a significant role at the different stages of tumorigenesis [118]. A downstream effector that was implicated in the GIF1/GIF1B-driven growth is LSD1. As expected, the use of LSD1 inhibitors, GSK-LSD1 and ORY-1001, attenuated the proliferation GIF1/GIF1B-driven medulloblastoma cells both in vitro and in mouse models with flank-implanted tumors, however, not those with intracranially implanted tumors (Figure 4). This indicates that these agents have inadequate brain–blood barrier penetration, which calls for evaluating other agents with better pharmacologic properties or alternative drug delivery methods [118]. Additionally, gene set enrichment analysis revealed an especially elevated activity of the folate metabolism pathway in group 3 medulloblastomas. Subsequently, the utility of the combination of the folate pathway inhibitor pemetrexed and the chemotherapeutic agent gemcitabine was investigated, and it showed significant inhibition of the growth of both in vitro and in vivo models of group 3 medulloblastoma. This combination is currently being investigated as part of a phase II clinical trial (NCT01878617) in the arms involving intermediate- and high-risk patients with non-WNT non-SHH medulloblastoma. Another pattern that was identified in group 3 medulloblastoma cells is the abundance of CD47 on their surface. CD47 is a cell membrane protein that helps cells evade being phagocytosed by cells of the innate immunity, and it achieves this effect by downstream activation of the signal regulatory protein alpha (SIRPα). In this framework, systemic treatment with Hu5F9-G4, which disrupts the interaction between CD47 and SIRPα, resulted in shrinkage of primary tumors and leptomeningeal metastasis in mouse models with implanted patient-derived group 3 medulloblastoma xenografts and mouse cell lines of this subgroup. On the other hand, intraventricular delivery of Hu5F9-G4 provided a more effective route in combating leptomeningeal and spinal metastases; however, it did not show significant impact at the primary tumor site [119]. This suggests that if this therapeutic agent reaches clinical application, the preferred mode of delivery might be dependent on the stage of the disease and the extent of resection of the primary tumor during surgery. Importantly, Hu5F9-G4 was determined to have minimal cytotoxic side effects on normal central nervous system cells, implying a favorable safety profile and a greater potential for clinical use [119]. In addition to targeting metastatic sites themselves, other experiments focused on hindering the initiation of metastasis altogether. Herein, the NOTCH1 pathway was implicated as an important contributor to the invasion and migration abilities of group 3 medulloblastoma cells. As a result, the intrathecal administration of anti-NOTCH1 Negative Regulatory Region antibody (anti-NRR1) in mice bearing group 3 medulloblastomas resulted in an attenuation of the metastatic potential of these tumors and a survival advantage for the treated mice [120]. Due to the lower response of group 3 medulloblastomas to conventional chemotherapeutic regimens, research has also focused on underscoring the mechanisms that mediate its chemoresistance and exploring druggable targets. In particular, the interleukin-6 (IL6)/gp130/Janus kinase (JAK)/signal transducer and activator of transcription 3 (STAT3) signaling pathway has been implicated in the development of chemoresistance to vincristine in group 3 medulloblastomas [121,122]. As a result, gp130 inhibitors (SC144 or bazedoxifene) and a JAK inhibitor (ruxolitinib) were individually investigated at non-cytotoxic doses and resulted in overcoming the acquired IL6-mediated resistance to vincristine in group 3 medulloblastomas [122].

The use of non-coding RNAs in the treatment of this subgroup has been widely studied with the hopes of finding alternative more effective therapies. Contextually, RNA-seq and miRNA profiling showed that medulloblastoma cells, especially those belonging to group 3, are enriched in miR-217 expression, which is known to promote proliferation and survival. As expected, interfering with miR-217 using anti-miR significantly reduced the proliferative, invasive, and migratory capacity of the group 3 medulloblastoma cell line HDMB03 [123]. Another miRNA that comprises a potential target is the pro-tumorigenic and pro-metastatic miR-183~96~182 complex, which was found to be upregulated in MYC-amplified medulloblastomas and to be an upstream inducer of PI3k/mTOR signaling [124]. On the other hand, several miRNAs were found to be downregulated in this subgroup, and their introduction can serve as a promising therapeutic avenue. In fact, the good prognostic role of two miRNAs, miR-148a and miR-193a, was discovered when group 3 medulloblastoma were compared to WNT medulloblastomas, which are known to carry the most favorable prognosis. In particular, restoration of miR-193a expression in group 3 medulloblastoma cells through treatment with 5-aza-deoxycytidine, a DNA methylation inhibitor, resulted in reduced proliferation and increased apoptosis and radiosensitivity of these cells [125]. In a similar fashion, miR-148a was found to be overexpressed in WNT medulloblastomas compared to the other subgroups, and rescuing its expression in D425 cells hindered their proliferative and invasive potential through downregulation of neuropillin1 (NRP1) [126]. In addition, two other miRNAs, miR-211 and miR-212-3p, were proven to elicit anti-oncogenic effects in in vitro models of group 3 medulloblastoma [127,128]. In addition to miRNAs, long non-coding RNAs (lncRNAs) also attracted the interest of researchers, and their expression profiles and implications in oncogenesis were investigated [129]. Herein, lnc-HLX-2-7 exhibited a differential overexpression in group 3 medulloblastomas, and its depletion resulted in attenuated proliferation and survival of cells belonging to this subgroup [130]. In fact, regulation of this lnRNA might be a major mechanism that mediates the previously discussed actions of the BRD4 inhibitor, JQ1, in this subgroup [130].

It important to note that the use of combination therapies has also emerged as an appealing route in the fight against group 3 medulloblastoma due to the aggressiveness and resistance of this subgroup. For instance, it has been shown that the combination of JQ1 with a CDK2 inhibitor (Milciclib), an mTOR inhibitor (BEZ235), or a pan-HDAC inhibitor (Panobinostat) resulted in synergistic anti-cancer effects in MYC-driven group 3 medulloblastoma and amplified the potential of each of the combined agents [131,132,133].

### 3.4. Group 4 Medulloblastoma

Despite its prevalence, this subtype has the poorest characterization when it comes to genetic and molecular profiles. It also lacks representative preclinical models that allow for better profiling of this group and the development of targeted therapies. In specific, the only currently available cell lines that are used for group 4 research are CHLA-01-MED and CHLA-01R-MED, which were derived from the same patient [134]. This lack of representative cell lines and the ambiguity of this group’s biology is reflected in the scarcity of selective therapies that target it. In fact, group 4 is frequently lumped with group 3 when it comes to the development of targeted therapies, even in emerging clinical trials. For instance, in the SJDAWN phase 1 clinical trial (NCT03434262) both group 3 and group 4 tumors belong to the same arm receiving ribociclib combined with gemcitabine although the efficacy of CDK4/6 inhibitors has only been proven in preclinical models of group 3 medulloblastoma. Nevertheless, tumors that belong to group 4 are also expected to respond to these inhibitors due to the high activity of CDK6 in this subgroup. In addition, it was suggested that LSD1 inhibitors have promising anti-proliferative effects against GFI1/GFI1B-driven group 3/4 medulloblastomas [118].

In light of the aforementioned findings, it is obvious that the development of adequate preclinical models for group 4 and the testing of novel agents that target it are crucial steps. Notably, some groups attempted to identify molecular targets that are overexpressed in this subtype using the two available cell lines. In particular, it was shown that the RNA-binding protein Musashi1 (MSI1) is specifically overexpressed in group 4 medulloblastoma, and it correlates with worse prognosis. Consequently, the use of an Msi1 inhibitor, luteolin, showed significant reduction in the proliferation of CHLA-01-MED and CHLA-01R-MED cells in vitro and augmented the effect of vincristine treatment [135]. Another effector that has been proven to be overexpressed in group 4 tumors and to be associated with more aggressive phenotypes is EZH2 [136]. In fact, using DZNep, an inhibitor of EZH2, was able to attenuate the proliferation of medulloblastoma cells in vitro [136]. This suggests that EZH2 inhibition can be of great potential in this subgroup; however, this needs to be further investigated using in vitro and in vivo models that specifically mimic group 4 medulloblastomas. Finally, a genome-wide gene enrichment analysis revealed several pathways that can provide tractable therapeutic targets in this subgroup. Herein, the NOTCH pathway was found to be overexpressed and to be closely related to prognosis [137]. In addition, the expression of NOTCH was also found to be associated with that of several immune-related effectors [137], hence suggesting a central role for this pathway in the survival of these tumors and their interaction with the immune system. In fact, several previous studies have highlighted the role of NOTCH signaling in regulating TAMs and mediating the immune resistance of other cancer types [138,139,140,141]. Interestingly, the gamma-secretase inhibitors (MK-0751, RO4929097), which hinder the cleavage of NOTCH intracellular domain (NICD) and its translocation to the nucleus to promote downstream effects, have entered clinical trials as potential therapeutic agents for refractory pediatric central nervous system tumors [142] (Figure 4). However, there are no dedicated studies that investigate the efficacy of NOTCH inhibition in preclinical models of group 4 medulloblastoma yet. In addition, other pharmacologic interventions that were predicted to be effective in this subgroup of medulloblastoma based on gene enrichment analyses are those that inhibit the estrogen-related receptor gamma (ESRRG), the JAK-STAT pathway, and members of the nucleotide biosynthesis pathway, such as dihydrofolate reductase [137]. At any rate, the efficacy of these modalities is also yet to be backed up by evidence from in vitro and in vivo experiments.

Despite the relatively low research interest in this subgroup, some attempts were made at characterizing the expression and roles of non-coding RNAs. In this context, the lnRNA SPRIGHTLY was found to be overexpressed in this subgroup. Additionally, the knockdown of SPRIGHTLY attenuated the proliferation of group 4 medulloblastoma cell lines and patient-derived xenografts both in vitro and in vivo [143]. This suggests that targeting this lnRNA might provide a compelling treatment avenue in group 4 medulloblastoma. Moreover, miR-592 was shown to play an oncogenic role in this subgroup through promoting mTOR and MAPK signaling [144]. However, it remains unclear if targeting this miRNA will produce favorable effects and whether it might offer a novel therapeutic approach.

## 4. Other Non-Specific Combination Therapies in Medulloblastoma

The concomitant deployment of therapeutic strategies could potentially offer a groundbreaking approach to treating medulloblastoma more effectively and aggressively, especially in the pediatric population. Indeed, combination therapy has yielded sustained and effective therapeutic solutions for other challenging cancer types. Here, we detail possible empiric combination regimens that may yield novel therapeutic strategies for medulloblastoma.

### 4.1. Targeting the PI3K/mTOR Pathway

Medulloblastomas and other malignant brain tumors are often associated with genetic mutations and epigenetic modifications that activate the PI3K/AKT/mTOR signaling pathway [145]. Activation of this pathway induces cell proliferation, migration, survival, metabolism, growth, and angiogenesis, thus making it a potential target for novel medulloblastoma combination therapeutics [146,147]. Several papers in the literature have investigated combination therapy involving this pathway as a pharmacologic target. The aberrant activation of the Hedgehog (HH) pathway along with the PI3K/mTOR pathway is frequently implicated in high-risk medulloblastoma. The roles of both HH and PI3K-mTOR signaling pathways have been linked to cancer “stem” cells, which can contribute to drug resistance in medulloblastoma. In this context it has been shown that vismodegib synergized well with BEZ235, a PI3K/mTOR dual inhibitor, to delay tumor growth both in vivo and in vitro [147]. Furthermore, this combination therapy sensitized cells to cisplatin, the current standard of care for patients diagnosed with medulloblastoma.

Moreover, some researchers investigated the combination of ribociclib with PI3K/mTOR inhibitors to investigate in vivo and in vitro efficacy [148]. However, their findings indicated that while molecular analysis displayed increased activity in vitro, this therapeutic strategy struggled to show in vivo enhanced survival or delay in tumor growth. Further investigations could show promising results on how CDK 4/6 inhibitors can be coupled with PI3K inhibitors to significantly improve the survival of medulloblastoma-bearing mice.

HDAC inhibitors are another potential therapeutic option for medulloblastomas and are particularly effective against established MYC-driven medulloblastoma cell lines and patient-derived xenografts [4]. HDAC inhibitors have been shown to upregulate expression of the FOXO1 tumor suppressor and they also work synergistically with PI3K/mTOR inhibitors to reduce tumor growth [149]. Activation of the PI3K/AKT/mTOR pathway results in phosphorylation of FOXO1 and prevents it from entering the nucleus [150], providing a mechanistic explanation for the enhanced anti-tumor effect of HDAC inhibitor and PI3K/mTOR inhibitor combination therapy [149,151].

### 4.2. Targeting Tyrosine Kinases

The receptor tyrosine kinase (RTK) family plays a crucial role in the development of medulloblastoma. Inappropriate activation of proteins such as EGFR, PDGFR, and cMET have been linked to the development and growth of medulloblastoma. Targeting RTK pathways has emerged as a promising approach for developing novel therapeutics against this type of tumor.

One unique therapeutic strategy aimed to leverage the role of epigenetics in medulloblastoma to develop a combination approach involving RTK inhibitors and HDAC/DNA methyltransferase (DNMT) inhibitors [152]. Specifically, epigenetic modifiers may enhance the expression of genes involved in tumor suppression and may synergistically work with RTK inhibitors that promote tumor growth. HDAC inhibitors such as 4-phenylbutyrate (4PB), suberoylanilide hydroxamic acid (SAHA), trichostatin A (TSA), and valproic acid (VPA) and DNMT inhibitors such as 5-azacytidine (5-AZA) and 5-aza-2′-deoxycytidine (5-AZA-CdR) were investigated [152]. These were combined with imatinib, a multi-kinase inhibitor. The study found that combining certain RTK inhibitors with certain epigenetic modifiers, such as SAHA and 5-AZA-CdR, resulted in a significant reduction in medulloblastoma cell growth compared to treatment with either agent alone. The combination of small-molecule inhibitors of RTKs and epigenetic modifiers may be a promising therapeutic approach for medulloblastoma. The potential of this therapeutic strategy was corroborated by the combination of 4PB with gefitinib/vandetanib and showed similar in vitro efficacy [153]. The researchers also observed changes in the expression of genes involved in cell proliferation and DNA damage response, suggesting that the combination of 4-PB and RTK inhibitors may have multiple targets in brain tumor cells [153]. Further research is needed to determine the optimal combination of agents and to assess the safety and efficacy of this approach in vivo [152,153].

The previously mentioned cMET is an RTK that is overexpressed in many human cancers, including medulloblastoma, while focal adhesion kinase (FAK) family members are involved in the regulation of cell adhesion, migration, and invasion. Previous literature has identified that cMET and FAK potentially cooperate in medulloblastoma. When co-overexpressed in medulloblastoma cells, the two proteins formed a complex and cooperated to promote cell proliferation, migration, and invasion [148,154]. RNA interference knockdown of cMET or FAK family members individually resulted in a partial reduction in medulloblastoma cell growth, while knockdown of both cMET and FAK family members resulted in a significant reduction in cell growth. A combination of cMET and FAK inhibitors resulted in a synergistic reduction in cell growth compared to treatment with either agent alone in vitro. However, since an oral FAK inhibitor was not pharmaceutically available, in vivo work utilizing this combination approach remains to be completed [148]. A recent study investigating glioblastoma showed that temozolomide and radiation treatment resulted in the cleavage of FAK, which was found to be mediated by caspase 3 [155]. Cleavage of FAK disrupted its activity and prevented its ability to promote cell invasion. Future work combining temozolomide/radiation with cMET inhibitors could be an interesting therapeutic strategy that could translate these in vitro findings to in vivo models.

### 4.3. Targeting Vascular Endothelial Growth Factor

Tumor cells commonly overexpress vascular endothelial growth factor (VEGF), which promotes angiogenesis and is associated with enhanced tumor invasiveness, metastasis, and growth [156]. Gao et al. reported that miRNA-210, which has previously been shown to regulate VEGF expression in other tumor environments [157], is also elevated in medulloblastomas and may influence metastasis via regulation of VEGF expression [158].

The SHH pathway plays a critical role in the development of the cerebellum and is frequently dysregulated in medulloblastoma [159]. A recent study published by Krushanov et al. conducted an integrated molecular analysis of adult SHH-activated medulloblastomas and identified two clinically relevant tumor subsets with distinct molecular features and prognoses. The authors found that VEGFA expression was significantly upregulated in one of the subsets and that high VEGFA expression was associated with poor clinical outcomes. These findings suggest that VEGFA may serve as a potent prognostic indicator and a potential therapeutic target for SHH-activated medulloblastomas [97]. In addition to VEGFA, emerging evidence suggests that hypoxia-inducible factor 1 alpha (HIF-1α) may also play a critical role in medulloblastoma pathogenesis. HIF-1α is a transcription factor that regulates various cellular processes in response to hypoxic conditions and is frequently overexpressed in solid tumors [160].

Since HIF-1α activates the expression of genes that promote angiogenesis (formation of new blood vessels), metabolic adaptation, and cell survival under hypoxic conditions. Inhibition of HIF-1α can disrupt these adaptive responses and can potentially induce cell death or sensitize cancer cells to other treatments. Combination of VEGFA and HIF-1α inhibitors could provide a synergistic combination therapy against medulloblastoma. While both targets have been individually investigated, very little research has explored the combination of these therapeutic strategies in medulloblastoma. Further research is necessary to evaluate the efficacy of this combination therapy in clinical trials and to identify biomarkers that can predict response to treatment.

### 4.4. Immunotherapy

Immunotherapy has been shown to be a promising approach for treating medulloblastoma, as it can induce long-lasting anti-tumor immune responses. Adoptive immunotherapy using chimeric antigen receptor (CAR) T-cells has shown encouraging results in treating medulloblastoma and other brain tumors.

Recent studies have explored the potential of combining chemotherapy with immunotherapy to enhance the anti-tumor effects of both modalities [161]. Gemcitabine has been shown to decrease the proliferation and viability of DAOY cells and also decreased the expression of stem-cell-related genes in these cells. Next, this therapy was combined with generated anti-tumor T-cells that had been exposed to DAOY medulloblastoma cell lines. The combination treatment resulted in a significant reduction in tumor growth compared to either treatment alone in a subcutaneous xenograft model.

Another strategy developed CAR T-cells that were pre-targeted to the EPHA2, HER2, and IL 13-α2 receptors that are uniquely expressed in medulloblastoma. This strategy was coupled with the methylation inhibitor azacytidine to understand the potential for chemo-immunotherapy in targeting medulloblastoma [162]. These therapies were delivered intrathecally into mouse models of group 3 medulloblastoma. This study not only showed evidence for the repeat local delivery of CAR-T cells into CSF spaces but also showed that the combination of chemo-immunotherapy with trivalent CAR T-cells exhibited the highest clinical efficacy in this murine model. Future work demonstrating how azacytidine synergizes with CAR T-cells to promote an anti-tumor effect could help optimize therapy and improve future combination regimens.

In addition to CAR T-cell therapy, the use of immunomodulating agents has attracted wide interest in oncology research. This interest has been ignited by the discovery that cancer cells induce the overexpression of immune checkpoint proteins, which attenuate the immune response, on tumor infiltrating lymphocytes. Hence, immune checkpoint inhibitors were investigated as anti-cancer agents, and they have shown considerable efficacy against various solid tumors. This inspired testing their efficacy in the context of CNS tumors, including medulloblastoma [163]. Specifically, preclinical experiments on immune checkpoint blockade in SHH and group 3 medulloblastoma mouse models showed a differential response between the two subgroups. Indeed, mice with group 3 medulloblastoma showed a significant survival benefit after treatment with anti-programmed death 1 (anti-PD1) antibodies compared to untreated mice, whereas mice with SHH-activated medulloblastoma did not show a significant response to the same therapy. This differential response to immune-modulatory agents can be attributed to the distinct baseline characteristics of the tumor microenvironment and immune infiltration profile of the two subtypes [164]. In particular, the high expression of PD-1 on the lymphocytes of group 3 tumor suggests a greater reliance on the immunosuppressive function of immune checkpoint proteins, and thus a greater susceptibility to agents blocking the actions of these proteins [164]. Nevertheless, when it comes to clinical outcomes, conflicting results have been reported by observational studies regarding the response of pediatric medulloblastoma to immune checkpoint blockade [165,166]. Hence, the results of the currently ongoing phase II clinical trials (NCT03585465, NCT03173950) evaluating the efficacy of nivolumab in recurrent CNS tumors should help resolve this debate and provide valuable insights regarding the translational reproducibility of preclinical findings. Moreover, the new generation of immune-modulating agents involves antibodies that are agonists to the costimulatory pathways of immune cells [163]. For instance, APX005M, which is a monoclonal antibody that activates the costimulatory surface protein CD40, is also being investigated as part of a phase I clinical trial including pediatric patients with recurrent or refractory CNS malignancies (NCT03389802).

## 5. Conclusions and Future Directions

This review highlighted how the development of medulloblastoma subtypes can be tracked back to neuronal progenitors in the developing cerebellum. It also elaborated on the unique genetic, epigenetic, transcriptional, and translational profiles of each group, which have been excessively investigated and utilized for the design of efficacious targeted therapies. Indeed, the development of such therapies has shown great promise for the future of medulloblastoma treatment, and several agents have already entered clinical trials and are in advanced stages of testing. Table 2 offers a summary of all registered clinical trials that investigate molecular-targeted therapies in medulloblastoma. However, despite the intriguing preclinical results that the studied agents have shown, treatment resistance is not uncommon. Therefore, with the rapid emergence of such resistance to the currently available targeted therapies and due to the long periods required for developing and testing novel agents, the future of medulloblastoma therapeutics should focus more on the upfront use of combination therapies. As shown in this review, combination therapies can hinder the development and progression of medulloblastoma at different stages, thus eliciting synergistic effects in tumor control and regression. At any rate, the current status quo in studying and testing combination therapies in medulloblastoma is still suboptimal, and further future efforts should be channeled into this avenue.

## Figures and Tables

**Figure 1 cancers-15-03889-f001:**
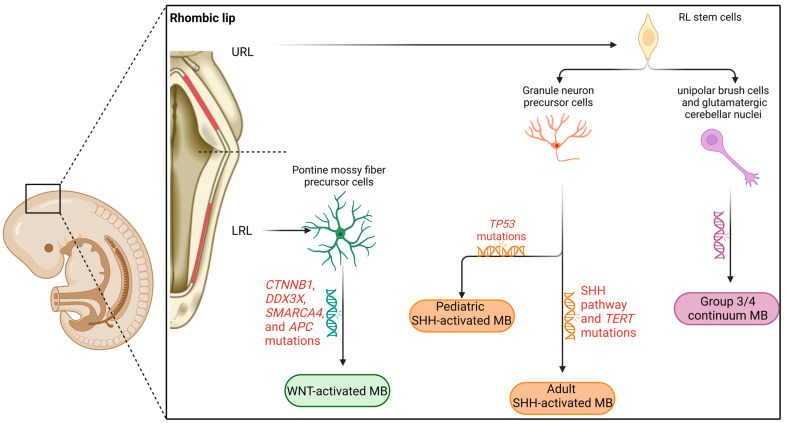
The neurodevelopmental origin of the four subgroups of medulloblastoma. WNT-activated tumors derive from the pontine mossy fiber precursor cell of the LRL and are characterized by mutations that involve the *CTNNB1*, *DDX3X*, SMARCA4, and *APC* genes. SHH-activated tumors derive from the granule neuron precursor cells of the URL, with the adult subtype commonly having mutations in the genes of the SHH pathway and TERT promoter and the pediatric subtype commonly possessing mutations in the TP53 gene. Finally, the unipolar brush cells and glutamatergic cerebellar nuclei give rise to the group 3/group 4 continuum of medulloblastoma tumors with a differential of transcriptional and DNA methylation profiles. LRL, lower rhombic lip; MB, medulloblastoma; RL, rhombic lip; SHH, sonic hedgehog; URL, upper rhombic lip; WNT, wingless.

**Figure 2 cancers-15-03889-f002:**
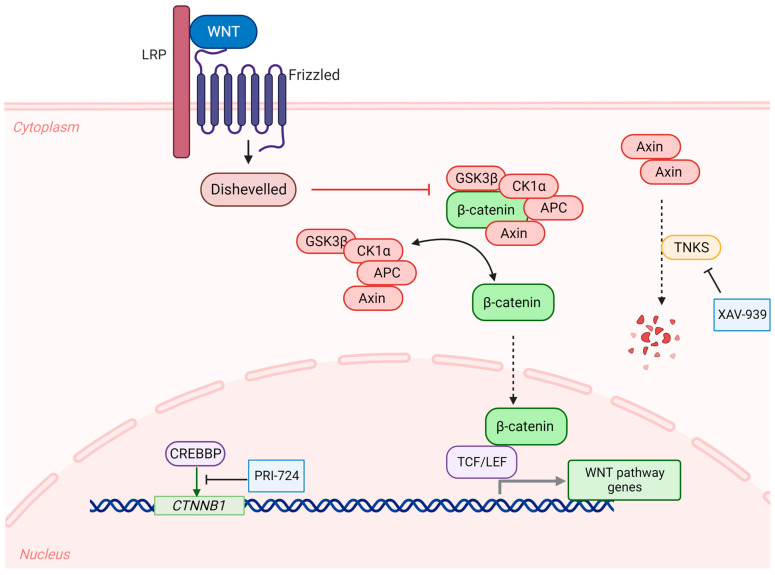
The activated canonical WNT pathway and potential pharmacotherapeutic options to target it. Canonical WNT signaling begins with the binding of the WNT ligand to the dimeric cell surface receptor composed of the Frizzled transmembrane protein and LRP. This causes the activation of the downstream Disheveled protein, which in its turn leads to the dissociation of β-catenin from the complex (Axin, APC, GSK3β, and CK1α) that tags it for degradation. The freed β-catenin can now translocate into the nucleus to cooperate with TCF/LEF in inducing the expression of the effector WNT pathway genes. Pharmacotherapeutic interventions that can target this pathway include PRI-724, which disrupts the CREBBP-mediated expression of the *CTNNB1* gene, which encodes β-catenin. Another drug is the TNKS inhibitor XAV-939, which can prevent the TNKS-mediated destruction of Axin, hence, leaving more Axin available to hinder the actions of β-catenin. APC, adenomatous polyposis coli; CK1α, Casein kinase 1 alpha; CREBBP, CREB binding protein; GSK3β, glycogen synthase kinase 3 beta; LRP, lipoprotein receptor-related protein; TCF/LEF, T-cell factor/lymphoid enhancer factor; TNKS, tankyrase; WNT, wingless.

**Figure 3 cancers-15-03889-f003:**
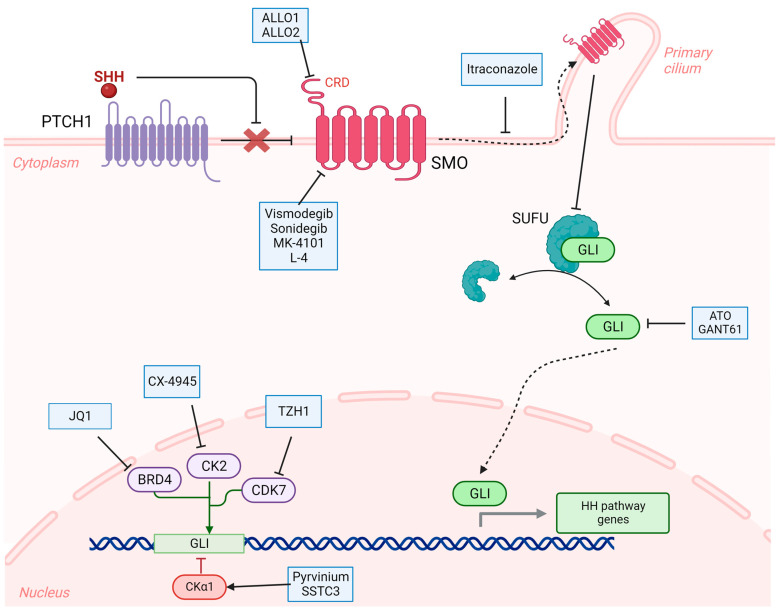
The activated SHH pathway and potential pharmacotherapeutic options to target it. SHH pathway signaling commences with the binding of the SHH ligand to the PTCH1 receptor. This binding lifts the PTCH1-mediated inhibition SMO. SMO is now able to translocate to the primary cilium where it can activate downstream signaling, mainly causing the dissociation of GLI from it repressor SUFU. The freed GLI protein translocates to the nucleus to induce the expression of effector SHH pathway genes. The expression of GLI itself is under the control of certain mediators, with BRD4, CK2, and CDK7 positively regulating the gene’s transcription and CKα1 negatively regulating it. Several pharmacotherapeutic agents can target the SHH pathway at different levels. Vismodegib, sonidegib, MK-4101, and L-4 can all inhibit SMO by binding to its transmembrane domain, while ALLO1 and ALLO2 can inhibit this receptor by binding to its CRD. On the other hand, itraconazole can block the actions of SMO by hindering its translocation to the primary cilium. Inhibiting the GLI protein can be achieved by directly targeting it via antagonists such as ATO and GANT61 or through targeting its expression. The latter process can be achieved by either inhibiting the GLI gene’s positive transcriptional regulators (BRD4, CK2, and CDK7) or activating its negative transcriptional regulator CK α1. ATO, arsenic trioxide; BRD4, bromodomain 4; CDK7, cyclin-dependent kinase 7; CKα1, casein kinase alpha 1; CK2, casein kinase 2; CRD, cysteine-rich domain; GANT61, GLI antagonist 61; GLI, glioma-associated oncogene; PTCH1, patched 1; SHH, sonic hedgehog; SMO, smoothened.

**Figure 4 cancers-15-03889-f004:**
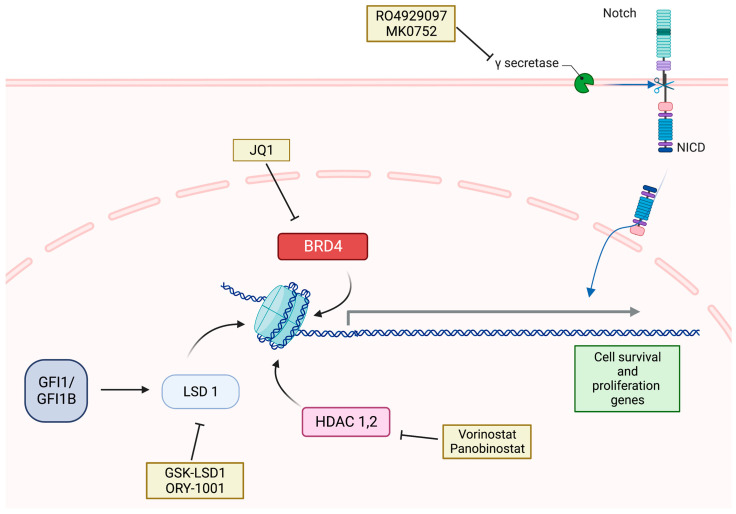
Molecular pathways implicated in tumoregenesis of non-WNT non-SHH medulloblastomas and potential pharmacotherapeutic options to target them. The NOTCH1 pathway is overactivated in group 3 and group 4 meduloblastoma. The activity of the pathway is dependent on the Gamma secretase cleavage of the NOTCH receptor, which results in the dissociation of NICD. Subsequently, NICD translocates to the nucleus and promotes the expression of downstream genes. In addition, the epigenetic modifiers HDACs 1/2, BRD4, and GFI1/GFI1B/LSD1 have been shown to play major roles in promoting the expression of pro-oncogenic genes (such as *MYC* in group 3 medulloblastoma). These epigenetic modifiers can be targeted using vorinostat and panobinostat for HDACs 1/2, JQ1 for BRD4, and GSK-LSD1 and ORY-1001 for LSD1. BRD4, Bromodomain 4; GFI, growth factor independent protein; HDAC, histone deacytylase; NICD, NOTCH intracellular domain.

**Table 1 cancers-15-03889-t001:** Clinical characteristics and the neurodevelopmental, genetic, and epigenetic profiles of the four molecular subgroups of medulloblastoma.

	WNT-Activated	SHH-Activated	Group 3	Group 4
Prevalence	10%	30%	25%	35%
5-year survival	>90%	70%	50%	75%
Neurodevelopmental origin	Pontine mossy fibers of the lower rhombic lip	Granule neuron precursor cells of the upper rhombic lip	Unipolar brush cells and glutamatergic cerebellar nuclei of the upper rhombic lip	
Commonly mutated genes	*CTNNB1*, *DDX3X*, *CREBBP*, *SMARC4*	*TP53*, *TERT*, *PTCH1*, *GLI2*, *SMO*, *SUFU*	*MYC*, *SOX11*, *PVT1*, *OTX2*, *GFI1/GFI1B*	*MYCN*, *SNCAIP*, *GFI1/GFI1B*
Important epigenetic players	ARID1, ARID2, SMARC4, promoter methylation of CDH1	MLL2/KMT2D, MLL3/KMT2C, NCOR2, LDB1	LSD1, PRC2, EZH2, BRD	KDM6A/UTX, LSD1

BRD, bromodomain; CREBBP, CREB binding protein; GFI, growth factor independent protein; GLI, glioma-associated oncogene; EZH, enhancer of zeste homolog; PRC, polycomb repressor complex; PTCH, patched; SHH, sonic hedgehog; SMO, smoothened; WNT, wingless. References for table [6,7,8].

**Table 2 cancers-15-03889-t002:** Summary of concluded and ongoing clinical trials investigating molecular-targeted agents in medulloblastoma.

Group	Agent	Mechanism of Action	Trials	Type/Design	Population	Intervention	Status
**SMO inhibitors**	Sonidegib (LDE-225)	Binds to the transmembrane portion of the SMO protein and inhibits downstream signaling	NCT04402073 (PersoMed-I)	Phase II Comparative Randomized	Adult and post-pubertal patients with SHH-activated medulloblastoma	Sonidegib and reduced dose radiotherapy	Recruiting
			NCT01708174	Phase II Single arm	Pediatric and adult patients with relapsed SHH-activated medulloblastoma	Sonidegib	Completed. Results available on ClinicalTrials.gov (accessed on 24 July 2023)
			NCT01208831	Phase I Dose escalation	Adult patients with advanced solid tumors (including medulloblastoma)	Sonidegib	Completed. Results available on Novartis website
			NCT01125800	Phase I/II Dose escalation	Pediatric and adult patients with recurrent or refractory SHH-activated medulloblastoma	Sonidegib	Completed. Results published [72]
			NCT00880308	Phase I Dose escalation	Adult patients with advanced solid tumors (including medulloblastoma)	Sonidegib	Completed. Results published [167]
	Vismodegib (GDC-0449)	Binds to the transmembrane portion of the SMO protein and inhibits downstream signaling	NCT01878617	Phase II Parallel assignment Non-randomized	Skeletally mature patients with newly diagnosed standard and high-risk SHH-activated medulloblastoma	Standard chemoradiotherapy with vesmodegib added to maintenance therapy	Active, not recruiting
			NCT01601184	Phase I/II Parallel assignment Randomized	Adult patients with recurrent or refractory SHH-activated medulloblastomas	Vismodegib plus temozolomide versus temozolomide alone	Terminated (number of successes not reached)
			NCT01208831 PBTC-032	Phase II Single group	Pediatric patients with recurrent or refractory medulloblastoma without (stratum A) or with (Stratum B) SHH activation	Vismodegib	Completed. Published results [71]
			NCT00939484 PBTC-025B	Phase II Single group	Adult patients with recurrent or refractory medulloblastoma without (stratum A) or with (Stratum B) SHH activation	Vismodegib	Completed. Published results [71]
			NCT00822458 PBTC-025	Phase I Dose finding	Young patients with recurrent or refractory medulloblastoma	Vismodegib	Completed. Published results [70]
	Taladegib (ENV-101)	Binds to the transmembrane portion of the SMO protein and inhibits downstream signaling	NCT05199584	Phase II Parallel assignment Randomized	Adult patients with refractory advanced solid tumors (including medulloblastoma) with loss of function mutations in the *PTCH1* gene	Taladegib	Recruiting
			NCT01697514	Phase I Single group	Pediatric patients with recurrent or refractory medulloblastoma or rhabdomyosarcoma	Taladegib	Withdrawn (poor recruitment)
	ZSP1602	SMO antagonist (specific mechanism not known)	NCT03734913	Phase I Parallel assignment Non-randomized	Adult patients with advanced solid tumors	ZSP1602	Unknown (last update in July 2020 was recruiting)
	LEQ506	Second generation SMO antagonist (specific mechanism not known) [168]	NCT01106508	Phase I Dose finding	Adult patients with advanced solid tumors	LEQ506	Completed. Results available on Novartis website
**GLI inhibitors**	ATO	Direct inhibitor of GLI	NCT00024258	Phase II Single group	Pediatric and adult patients with neuroblastoma and other pediatric solid tumors (nonmyeloid and nonlymphoid)	ATO	Completed. Results available on ClinicalTraial.gov (accessed on 24 July 2023)
	Silmitasertib (CX-4945)	CK2 antagonist that reduces the transcription of *GLI* genes	NCT03904862	Phase I/II Parallel assignment Non-randomized	Skeletally immature (phase I) and skeletally mature (phase II) patients with recurrent SHH-activated medulloblastomas	Silmitasertib with or without surgical resection	Recruiting
**HDAC inhibitors**	Vorinostat	Inhibitor of class I and II HDACs	NCT01076530	Phase I Single group	Young patients with relapsed or refractory primary CNS tumors	Vorinostat plus temozolomide	Completed. Published results [169]
			NCT00994500	Phase I Single group	Young patients with refractory or recurrent solid tumors (including medulloblastoma)	Vorinostat and Bortezomib (ubiquitin-proteosome pathway inhibitor)	Completed. Published results [170]
			NCT00867178	Phase I Single group	Younger patients with newly diagnosed CNS embryonal tumors	Adding vorinostat and isotretinoin to induction chemotherapy (cisplatin, etoposide, vincristine, cyclophosphamide)	Completed. Published results [171]
			NCT00217412	Phase I Parallel assignment Non-randomized	Young patients with recurrent or refractory solid tumors (including medulloblastoma), lymphoma, or leukemia	Vorinostat plus isotretinoin	Completed. Published results [172]
	Panobinostat (MTX110)	Pan-HDAC inhibitor	NCT04315064	Phase I Single group	Pediatric and adult patients with recurrent medulloblastoma	Infusions of Panobinostat into the fourth ventricle of the brain or tumor resection cavity	Recruiting
	Fimepinostat	Pan-HDAC and PI3K inhibitor	NCT03893487 PNOC016	Phase I Single group	Pediatric and adult patients with newly diagnosed DIPG, recurrent medulloblastoma (any subtype), or recurrent high-grade glioma	Fimepinostat 2 days preoperatively followed by surgical resection, then maintenance with fimepinostat	Active, not recruiting
	Romidepsin (FR901228)	HDAC inhibitor	NCT00053963	Phase I Single group	Pediatric patients with refractory or recurrent solid tumors	Romidepsin	Completed. Results not available
**Cell cycle-disrupting agents**	Prexasertib (LY2606368)	Checkpoint kinases 1 and 2 (CHK1/2) inhibitor	NCT04023669 (St. Jude ELIOT)	Phase I Parallel assignment Non-randomized	Pediatric and adult (up to 24 years old) patients with refractory or recurrent SHH-activated, group 3, or group 4 medulloblastoma	Prexasertib in combination with cyclophosphamide (all three subtypes) or gemcitabine (only groups 3 and 4)	Active, not recruiting
	Palbociclib	CDK4/6 inhibitor	NCT03709680	Phase I-Dose escalation Phase II-Randomized	Pediatric patients with refractory or recurrent solid tumors (including medulloblastoma)	Palboociclib combined with chemotherapy (temozolomide plus irinotecan or topotecan plus cyclophosphamide)	Recruiting
			NCT03526250 (Subprotocol of the NCI-COG Pediatric MATCH trial)	Phase II Single group	Pediatric patients with relapsed or refractory Rb-positive solid tumors non-Hodgkin lymphoma, or histiocytic disorders with activating alterations in cell cycle genes	Palbociclib	Active, not recruiting
			NCT02255461 (PBTC-042)	Phase I Single group	Pediatric patients with Rb-positive recurrent, progressive, or refractory primary CNS tumors.	Palbociclib	Completed. Published results [108]
	Ribociclib (LEE011)	CDK4/6 inhibitor	NCT05429502	Phase I/II Parallel assignment Randomized	Pediatric patients with relapsed or refractory solid tumors	Ribociclib combined with topotecan and temozolomide	Recruiting
			NCT03434262 (SJDAWN)	Phase I Parallel assignment Non-randomized	Pediatric and adult patients with refractory or recurrent brain tumors	Stratum A: ribociclib and gemcitabine for patients with recurrent/refractory group 3/4 medulloblastoma or ependymoma Stratum B: ribociclib and trametinib for recurrent/refractory WNT-activated or SHH-activated medulloblastoma and other CNS tumors Stratum C: ribociclib and sonidegib for skeletally mature patients with recurrent/refractory SHH-activated medulloblastoma	Active, not recruiting
			NCT03387020	Phase I Single group	Pediatric patients with recurrent, progressive, or refractory CNS tumors	Ribociclib and everolimus (mTOR inhibitor)	Completed. Published results [173]
	Abemaciclib	CDK4/6 inhibitor	NCT04238819	Phase I Dose escalation	Pediatric patients with recurrent or refractory solid tumors	Abemaciclib combined with temozolomide alone or with irinotecan and temozolomide	Recruiting
**Tyrosine kinase inhibitors (TKIs)**	Apatinib	TKI that blocks the activity of vascular endothelial growth factor receptor 2 (VEGFR2)	NCT04501718	Phase II Single group	Pediatric patients with recurrent medulloblastoma	Apatinib combined with temozolomide and etoposide	Recruiting
	Volitinib	TKI that blocks cMET signaling	NCT03598244	Phase I Single group	Pediatric patients with refractory, progressive, or recurrent primary CNS tumors	Volitinib	Recruiting
	Erdafitinib	TKI that blocks fibroblast growth factor receptor	NCT03210714 (Subprotocol of the NCI-COG Pediatric MATCH trial)	Phase II Single group	Pediatric patients with relapsed or refractory solid tumors non-Hodgkin lymphoma, or histiocytic disorders with FGFR mutations	Erdafitinib	Active, not recruiting
	Entrectinib (Rxdx-101)	TKI that blocks the activity of tropomyosin receptor kinases, ROS1, and ALK	NCT02650401	Phase I/II Single group	Pediatric patients with locally advanced, metastatic, or refractory solid or primary CNS tumors	Entrectinib	Active, not recruiting
	Adavosertib (MK-1775)	TKI that block the activity of WEE1	NCT02095132	Phase I/II Single group	Pediatric patients with relapsed or refractory solid tumors	Adavosertib combined with irinotecan	Active, not recruiting
	Cediranib (AZD-2171)	TKI that blocks the activity of VEGF	NCT00326664	Phase I Single group	Pediatric patients with recurrent, progressive, or refractory primary CNS tumors	Cediranib	Completed. Published results [174]
	Lapatinib	Dual TKI that blocks epidermal growth factor receptor and HER2 signaling	NCT00095940	Phase I/II Single group	Pediatric patients with recurrent or refractory CNS tumors	Lapatinib	Completed. Results available on ClinicalTrials.gov (accessed on 24 July 2023)
**Antiangiogenic factors**	Pomalidomide	Decreases the concentrations of VEGF and HIF1α. Increases the production of immune-stimulatory cytokines	NCT03257631	Phase II Single group	Pediatric patients with recurrent or progressive primary brain tumors	Pomalidomide	Completed. Published results [175]
	Bevacizumab and other drugs (multidrug)	Bevacizumab is a monoclonal antibody that binds VEGF. Thalidomide, celecoxib, and fenofibrate also have antiangiogenic effects [176]	NCT01356290	Phase II Single group	Pediatric patients with recurrent or progressive medulloblastoma, ependymoma, or ATRT.	Bevacizumab in combination with 5 oral drugs (thalidomide, celecoxib, fenofibrate, etoposide, and cyclophosphamide)	Recruiting
	PTC-299	Targets VEGF mRNA and inhibits their translation	NCT01158300	Phase I Single group	Pediatric patients with recurrent or refractory primary CNS tumors	PTC-299	Completed. Published results [177]
	Cilengitide	Integrin antagonist that disrupts endothelial interactions	NCT00063973 PBTC-012	Phase I Single group	Pediatric patients with refractory primary brain tumors	Cilengitide	Completed. Published results [178]
**Immunomodulatory agents**	Nivolumab	Monoclonal antibody against the immune checkpoint protein programmed death 1 (PD1)	NCT03585465	Phase I/II Parallel assignment Randomized	Pediatric patients with relapsed or refractory solid tumors	Nivolumab combined with cyclophosphamide and vinblastine (Arm A), capecitabine (Arm B), or metronomic chemotherapy (Metronomic+ Nivolumab arm)	Recruiting
			NCT03173950	Phase II Parallel assignment Non-randomized	Adult patients with recurrent select rare CNS cancers (including medulloblastoma)	Nivolumab	Recruiting
	Pembrolizamab	Monoclonal antibody against the immune checkpoint protein PD1	NCT02359565	Phase I Single group	Pediatric patients with recurrent, progressive, or refractory high-grade gliomas, DIPGs, hypermutated brain tumors, ependymoma, or medulloblastoma	Pembrolizumab	Recruiting
	Cemiplimab (REGN2810)	Monoclonal antibody against the immune checkpoint protein PD1	NCT03690869	Phase I	Pediatric patients with relapsed or refractory solid or CNS tumors	Cemiplimab	Recruiting
	Indoximod	Inhibitor of the immune-suppressive enzyme Indoleamine-2,3-dioxygenase (IDO)	NCT05106296	Phase I Single group	Patients aged 12–25 years with pediatric brain tumors	Indoximod combined with ibrutinib (Bruton’s tyrosine kisase inhibitor, and chemoradiotherapy	Recruiting
			NCT04049669	Phase II Crossover Non-randomized	Pediatric patients with relapsed brain tumors or newly diagnosed DIPG	Indoximod administered during chemotherapy and/or radiation therapy	Recruiting
			NCT02502708	Phase I Parallel assignment Non-randomized	Pediatric patients with progressive primary brain tumors	Indoximod in combination with temozolomide-based chemotherapy	Completed. No results available
	Sotigalimab (APX005M)	CD40 agonist that activates antigen-presenting cells	NCT03389802	Phase I Sequential Non-randomized	Pediatric patients with recurrent, progressive, or refractory primary malignant CNS tumor	Sotigalimab	Active, not recruiting
**EZH2 inhibitors**	Tazemostat	EZH2 inhibitor	NCT03213665 (Subprotocol of the NCI-COG Pediatric MATCH trial)	Phase II Single group	Pediatric patients with relapsed or refractory solid tumors, non-Hodgkin lymphoma, or histiocytic disorders with gain of function mutations in EZH2 or loss of function mutations in SMARCB1 or SMARCA4	Tazemostat	Active, not recruiting
**PI3K/mTOR inhibitors**	Samotolisib (LY3023414)	Dual PI3K and mTOR inhibitor	NCT03213678 (Subprotocol of the NCI-COG Pediatric MATCH trial)	Phase II Single group	Pediatric patients with relapsed or refractory solid tumors, non-Hodgkin lymphoma, or histiocytic disorders with TSC loss of function mutations, and/or other PI3K/mTOR activating mutations	Samotolisib	Recruiting
	Sirolimus	mTOR inhibitor	NCT02574728	Phase II Single group	Pediatric patients with relapsed or refractory solid or CNS tumors	Sirolimus in combination with metronomic chemotherapy	Recruiting
**BRD inhibitors**	BMS-986158 and BMS-986378	BRD inhibitors that prevent the interaction between BET proteins and histones	NCT03936465	Phase I Parallel assignment Non-randomized	Pediatric patients with relapsed or progressive solid or CNS tumors	BMS-986158 or BMS-986378 as monotherapies	Recruiting
**Gamma secretase inhibitors**	RO492909 7	Blocks the cleavage of Notch intracellular domain (NICD) and its translocation to the nucleus to induce the expression of Notch pathway effector genes	NCT01088763	Phase I/II Single group	Pediatric patients with relapsed or refractory solid tumors, CNS tumors, lymphoma, or T-cell leukemia	RO4929097	Terminated
	MK0752	Blocks the cleavage of Notch intracellular domain (NICD) and its translocation to the nucleus to induce the expression of Notch pathway effector genes	NCT00572182	Phase I Single group	Pediatric patients with recurrent or refractory CNS tumors	MK0752	Terminated due to discontinued financial support
**JAK/STAT inhibitors**	WP1066	JAK2/STAT3 pathway inhibitor	NCT04334863	Phase I Single group	Pediatric patients with recurrent or progressive malignant brain tumors	WP1066	Completed. No results available
**Others**	TB-403	Monoclonal antibody against placental growth factor (PIGF)	NCT02748135	Phase I Single group	Pediatric patients with relapsed or refractory medulloblastoma, neuroblastoma, Ewing sarcoma, and alveolar rhabdomyosarcoma	TB-403	Completed. Published results [179]
	Mebendazole	Antiparasitic drug that has been shown to have anti-proliferative and proapoptotic roles in several cancer types via its ability to modulate several oncogenic pathways (including SHH, MEK/ERK, and STAT1/2)	NCT02644291	Phase I Single group	Pediatric patients with recurrent or progressive brain tumors	Mebendazole	Completed. No results available

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
