# Peer review of "The Neurodevelopmental and Molecular Landscape of Medulloblastoma Subgroups: Current Targets and the Potential for Combined Therapies"

_cancers, 2023, doi:10.3390/cancers15153889_

Round 1
Reviewer 1 Report
1. Is each subtype related to current clinical treatment? For example, is the ICB beneficial for any specific subtype? If so, why?
2. On page 11, SMO D743H mutation is a typo or mistake should be D473H.
3. The AMPK-mTOR signaling regulates GLI1, but the author did not address this in detail.
Author Response
Response to the reviewers:
We thank the reviewers for taking the time to go over the manuscript and for their valuable feedback. Below are our responses to their comments. The corresponding changes made to the text are highlighted in yellow in the manuscript file.
Reviewer 1:
- Is each subtype related to current clinical treatment? For example, is the ICB beneficial for any specific subtype? If so, why?
Currently, there are no major differences in the standard of care for the four different medulloblastoma subgroups. As mentioned in the introduction, the clinical treatment for all cases of medulloblastoma is maximal safe resection, craniospinal irradiation with a boost to the tumor bed, and adjuvant chemotherapy. Great efforts are being put forward by researchers in the field to investigate novel subtype-specific therapeutic agents, which were explored in our review. In this context, the reviewer suggests a very important point regarding the use of immune checkpoint blockers (ICBs) in medulloblastoma. Some ICBs were mentioned in the table summarizing the agents that are currently being investigated in clinical trials. Nevertheless, we agree with the reviewer that the use of these agents should be further elaborated on in the text. So, we added a paragraph elaborating on the evidence that is available on the efficacy of ICBs in medulloblastoma. In addition, as suggested by the reviewer, we explored the differential response to ICBs among subtypes of medulloblastoma and mentioned the underlying possible reasons for this differential response. The added paragraph is present in the subsection “4.4. Immunotherapy” (lines 1012 till 1039).
- On page 11, SMO D743H mutation is a typo or mistake should be D473H.
Thank you for pointing this typo out. We have fixed it to become D473H (line 505).
- The AMPK-mTOR signaling regulates GLI1, but the author did not address this in detail.
Thank you for bringing this topic to our attention. Indeed, an interesting interaction exists between AMPK, mTOR, and GLI1, which has been explored as a therapeutic target in other types of cancer and is worth being highlighted in the context of medulloblastoma. As per the reviewer’s suggestion, we have addressed the AMPK-mediated regulation of GLI1, both directly and indirectly (through mTOR). The added information can be found in the subsection “3.2. SHH-activated medulloblastoma” (lines 574 till 589).

Reviewer 2 Report
In this article, the authors summarized and reviewed the current knowledges on the different profiles of medulloblastoma subtypes, elaborate on the pharmacologic therapies that have been investigated to target each, and suggest potential combination therapies which is considered to conduct superior outcomes.
# Comments:
1) The authors should show more information by the tables and figures; at least, the current clinical molecular gradings of medulloblastomas and the graphical summary of signaling pathways which regulates medulloblastoma oncogenicity other than WNT pathway and SHH pathway together with the molecular target drugs related to these pathways should be additionally demonstrated as the tables or figures.
2) The authors should also review and summarize the epigenetic regulatory mechanisms which are considered as important for current molecular diagnosis of medulloblastomas and the drugs targeting these machineries by both text and figures.
Author Response
Response to the reviewers:
We thank the reviewers for taking the time to go over the manuscript and for their valuable feedback. Below are our responses to their comments. The corresponding changes made to the text are highlighted in yellow in the manuscript file.
Reviewer 2:
In this article, the authors summarized and reviewed the current knowledges on the different profiles of medulloblastoma subtypes, elaborate on the pharmacologic therapies that have been investigated to target each, and suggest potential combination therapies which is considered to conduct superior outcomes.
- The authors should show more information by the tables and figures; at least, the current clinical molecular gradings of medulloblastomas and the graphical summary of signaling pathways which regulates medulloblastoma oncogenicity other than WNT pathway and SHH pathway together with the molecular target drugs related to these pathways should be additionally demonstrated as the tables or figures.
Thank you for your suggestion. We have added a table (Table 1 in the updated manuscript) which summarizes the characteristics of each of the four medulloblastoma molecular subtypes. Moreover, we have added a new figure (Figure 4) that shows additional molecular pathways that are involved in the tumorigenesis of non-WNT non-SHH medulloblastomas. In addition, we indicated, in the figure, the drugs that can target each of these pathways. We believe that the additions suggested by the reviewer make the paper more appealing and easier for the readers to comprehend.
- The authors should also review and summarize the epigenetic regulatory mechanisms which are considered as important for current molecular diagnosis of medulloblastomas and the drugs targeting these machineries by both text and figures.
Thank you for highlighting this point. Indeed, the topic of epigenetic regulatory mechanisms is an important one to address. So, we added a paragraph at the end of each of the subsections “2.2. WNT-activated medulloblastoma” (lines 146 till 152), “2.3. SHH-activated medulloblastoma” (lines 189 till 195), “2.4. Group 3 medulloblastoma” (lines 241 till 258), and “2.5. Group 4 medulloblastoma” (lines 287 till 297) detailing the epigenetic mechanisms that are implicated in the development of each subtype. We also summarized the most important epigenetic factors in the added Table 1. As for drug targeting, we made sure that the new figure (Figure 4) contains a representation of the epigenetic targets for group 3 and group 4 that were already mentioned in the text (HDAC inhibitors, BRD4 inhibitor, and GFI/LSD inhibitors). Moreover, targeting the epigenetic factor BRD4 has already been illustrated in Figure 3 for SHH-activated medulloblastoma.

Round 2
Reviewer 1 Report
The comments are well taken care of, I approve the revision for publication.
Reviewer 2 Report
I consider the authors responded to all my requests appropriately.